# Understanding Metric Learning on Unit Hypersphere and Generating Better Examples for Adversarial Training

## Abstract

Recent works have shown that the adversarial examples can improve the performance of representation learning tasks. In this paper, we boost the performance of deep metric learning (DML) models with adversarial examples generated by attacking two new objective functions: *intra-class alignment* and *hyperspherical uniformity*. These two new objectives are motivated by our theoretical and empirical analysis of the tuple-based metric losses on the hyperspherical embedding space. Our analytical results reveal that a) the metric losses on positive sample pairs are related to intra-class alignment; b) the metric losses on negative sample pairs serve as uniformity regularization on hypersphere. Based on our new understanding on the DML models, we propose **A**dversarial **D**eep **M**etric **L**earning model with adversarial samples generated by **A**lignment or **U**niformity objective (ADML+A or U). With the same network structure and training settings, our ADML+A and ADML+U consistently outperform the state-of-the-art vanilla DML models and the baseline model, adversarial DML model with attacking triplet objective function, on four metric learning benchmark datasets.

## 1 Introduction

Deep metric learning (DML) has been applied to various computer vision tasks ranging from face recognition (Schroff et al., 2015; Liu et al., 2017) to zero-shot learning (Romera-Paredes & Torr, 2015; Bucher et al., 2016) and image retrieval (Song et al., 2016; Wu et al., 2017). It has been proved to be one of the most effective methods for learning the distance-preserving features of images. The intuition of DML is to pull the embedding of positive images pairs together and push the negative pairs apart, where the embedding function could be a deep neural network. Most of the metric losses in DML are tuple-based (Schroff et al., 2015; Song et al., 2016; Wu et al., 2017; Wang et al., 2019) or classification-based (Movshovitz-Attias et al., 2017; Kim et al., 2020; Boudiaf et al., 2020), these different losses have been shown to achieve similar performance in the recent reviews of DML (Roth et al., 2020; Musgrave et al., 2020).

One common ground of existing DML models is that the embedding space is a unit hypersphere. It is widely known that achieving uniformity on hypersphere can increase the generalization of models and preserve as much information as possible (Bachman et al., 2019; Liu et al., 2018; 2021; Hjelm et al., 2018), and the objective function that lead to uniformity is called uniformity regularization. Meanwhile, the downstream tasks in DML favor the models with small intra-class alignment (Wu et al., 2017; Wang et al., 2019). In this work, we investigate these two properties, *intra-class alignment* and *hyperspherical uniformity* (Wang & Isola, 2020) for tuple-based metric losses. We derive the theoretical analysis for the triplet loss to prove that the triplet loss on the positive sample pairs minimizes the *intra-class alignment* by mapping all samples from one class to the same vector, while the triplet loss on the negative sample pairs achieves *hyperspherical uniformity*. We further conduct empirical studies to show that the same statement is also valid for other tuple-based metric losses.

We utilize our new understanding on DML to design novel robust DML methods to enhance the performance via improved adversarial training. Adversarial training aims at improving the robustness of models towards to certain types of attacks by training with perturbed samples. However, as shown in the recent work on contrastive representation learning (Jiang et al., 2020), adversarial training can also enhance the natural performance on the downstream classification task. Due to

the similarity between contrastive learning and deep metric learning, we believe it's also possible to improve the nature performance of metric learning models with adversarial samples. Following our new insights on positive and negative metric losses, we generate perturbations by attacking the alignment or uniformity objective, and create adversarial DML models augmented with both normal samples and perturbed samples. Our experimental results show that the new adversarial DML models can significantly boost the natural performance.

The major contributions of our paper can be summarized as follows:

- We analyze the intra-class alignment and hyperspherical uniformity for tuple-based metric losses, and establish the connections between these two properties and the positive/negative metric losses.
- Based on our new analysis and understanding, we propose two new adversarial DML models, ADML+A and ADML+U, via attacking the alignment or uniformity objective. ADML+A and ADML+U improve the natural performance on benchmarks significantly.

## 2 RELATED WORKS

**Deep metric learning.** There are mainly two kind of metric losses in DML, tuple-based and classification based losses. Tuple-based losses include contrastive loss (Hadsell et al., 2006), triplet loss (Schroff et al., 2015), margin loss (Wu et al., 2017), and multi-similarity loss (Wang et al., 2019), where the objective function is based on the distance between positive pairs and negative pairs. In classification-based losses, the learning objective is not depend on the positive or negative pairs but a fixed (Boudiaf et al., 2020) or learnable proxy (Kim et al., 2020). In the recent reviews of metric learning methods (Roth et al., 2020; Musgrave et al., 2020), it's concluded the improvement on the DML performance is mainly due to different training strategies and unfair comparison. The original contrastive loss and triplet loss still achieved comparable result with other metric losses under the same network and training strategies. In experiments we apply the training framework of (Roth et al., 2020) to ensure fair comparison.

**Learning with hyperspherical uniformity.** Hyperspherical learning regards learning tasks where the embedding space is a unit hypersphere. The uniformity of the hypersphere represents the diversity of vectors on the sphere. It encourages vectors to be spaced with angles as large as possible so that these vectors can be evenly distributed on the hypersphere (Liu et al., 2018). Achieving hyperspherical uniformity can help with preventing overfitting and improving generalization of the neural works (Liu et al., 2021). The objective functions which can lead to the uniformity on hypersphere are called uniformity regularization. Hyperspherical embedding is widely applied in representation learning tasks such as contrastive representation learning (Oord et al., 2018; Hjelm et al., 2018) and DML (Wu et al., 2017; Liu et al., 2017). Wang & Isola (2020) showed that the objective function in contrastive representation learning optimizes for intra-class alignment and uniformity together.

**Adversarial examples improves natural performance.** In classification tasks, it is well known that the clean accuracy of adversarially trained model is typically worse than the normal model. However, Xie et al. (2020) showed that adversarial samples can be used to improve the clean accuracy of image classification models. According to (Jiang et al., 2020), training with adversarial samples can help improve the natural performance of contrastive learning models on the downstream classification tasks. The authors presented adversarial attacks based on the objective of contrastive learning and achieved improvement on both natural and robustness performance. It's believed that adversarial examples contain extra features, thus the generalization of models augmented with adversarial example is increased (Ilyas et al., 2019; Salman et al., 2020; Xie et al., 2020), which contributes to better natural performance. Duan et al. (2018) utilized the triplet loss as the attacking objective to generate adversarial examples to improve DML models. To our best knowledge, this is the only one work using adversarial example to boost the natural performance of DML. In our work, we show that alignment and uniformity loss can generate stronger adversarial examples comparing to triplet loss, and thus lead to better generalization of DML models.

## 3 ALIGNMENT AND HYPERSPHERICAL UNIFORMITY IN TUPLE-BASED METRIC LOSSES

In this section, we will study tuple-based metric losses on the unit hypersphere embedding space. We assume having $n$ classes $X_1, \cdots, X_n$ in training set and denote the encoder by $f : \mathbb{R}^d \to \mathcal{S}^{k-1}$

where $\mathcal{S}^{k-1}$ is the surface of a $k$-dimensional unit ball. Let $p_{data}(\cdot)$ be the data distribution over $\mathbb{R}^d$, $p_{pos}(\cdot, \cdot)$ be the distribution of positive pairs over $\mathbb{R}^d \times \mathbb{R}^d$, and $p_{tri}(\cdot, \cdot, \cdot)$ be the distribution of triplet pairs over $\mathbb{R}^d \times \mathbb{R}^d \times \mathbb{R}^d$, where the first two entries have the same label and the third entry is a sample from different classes. Please note that all detailed proofs are included in supplementary material Appendix E. We also conduct experiments to validate our theoretical analysis, the details is in Appendix C.

The major intuition of DML is to pull the representations of similar samples together and push dissimilar samples apart. Thus, we reformulate the metric losses as the combination of two parts:

- **Positive metric loss**: minimizes the distance between embedded positive sample pairs.
- **Negative metric loss**: maximizes the distance between embedded negative sample pairs.

Although in DML models the positive metric losses have different representations, they share one common optimal solution pattern, where samples from the same class are mapped to the same feature vector. Thus, we define the alignment loss with minimizing the intra-class distance.

**Definition 1.** *(Intra-class alignment) The expectation of intra-class distance is given by:*

$$\mathcal{L}_{alignment}(f; X, p_{pos}) := \mathbb{E}_{(x,y)\sim p_{pos}} \left[ ||f(x) - f(y)||_2^2 \right] \tag{1}$$

*the minimum of this loss is achieved when the samples with the same label are encoded to the same embedding.*

**Proposition 1.** *If the support set of the data distribution is connected and the support set of each class distribution is closed, the minimum of $\mathcal{L}_{alignment}$ is reached when **all** samples are projected to the same vector.*

In Sec. C.1, we conduct the empirical studies to verify our analysis. Results in Table 8 show that samples are roughly projected to the same vector if only the positive metric losses are used.

The negative metric losses aim at positioning the embedding of dissimilar samples as far as possible. However, because the embedding space of DML is a unit hypersphere, where the maximum distance between two points is 2, it's not possible to separate all negative embeddings with a large margin. Actually on $\mathcal{S}^{k-1}$, the number of points with pairwise distance larger or equal than $\sqrt{2}$ is at most $2k$ and the embedding dimension $k$ is always much smaller than the number of feature vectors, thus it's impossible to make all distances between negative pairs exceed $\sqrt{2}$. Therefore investigating the properties of negative metric losses on the unit hypersphere is an interesting and important topic. We believe the negative metric losses are closely related to the uniformity on the hypersphere and our experimental results support this argument.

**Definition 2.** *(Hyperspherical uniformity) The embedded samples should be evenly distributed on the spherical surface.*

In practice, the hyperspherical uniformity can be achieved by optimizing the uniformity regularization. There exist many different representations of the regularization, and we utilize the hyperspherical energy (HE) (Liu et al., 2018):

$$E(s, X) = \begin{cases} \mathbb{E}_{x\sim p_{data}, y\sim p_{data}}[||f(x) - f(y)||_2^{-s} 1_{x\neq y}], s > 0 \\ \mathbb{E}_{x\sim p_{data}, y\sim p_{data}}[\log(||f(x) - f(y)||_2^{-1} 1_{x\neq y})], s = 0 \end{cases} \tag{2}$$

and Gaussian hyperspherical energy (G-HE) (Wang & Isola, 2020):

$$E_G(s, X) = \log \mathbb{E}_{x\sim p_{data}, y\sim p_{data}}[e^{-s||f(x)-f(y)||_2^2}], s > 0 \tag{3}$$

in the experiments for comparison. The values of HE and G-HE can also be used as measurements on the uniformity of the embedded samples. We expect the value to be small in order to achieve good hyperspherical uniformity. We also want to mention that simply maximizing the distance between samples will not lead to hyperspherical uniformity, and the detailed discussion is in Sec. B.2.

Because finding the optimal solution of the HE or G-HE problem is NP-hard (Liu et al., 2018), we are not able to calculate the exact position of vectors which are evenly distributed on the sphere. We provide a primary insight about how should finite vectors be uniformly distributed on unit hypersphere, and our conclusion is consist with the empirical results.

Since there exist many different tuple-based metric losses, analyzing all of them theoretically is impossible in this work. In Sec. 3.1, we will provide the theoretical analysis of the triplet loss. The analysis of linear loss can be found in Sec. B.2. In Appendix C we will show the empirical results on four popularly used tuple-based metric losses to verify our statement.

## 3.1 TRIPLET METRIC LOSSES

In this subsection, we provide our theoretical analysis on the triplet metric losses with the following assumptions.

**Assumption.** Distributions $p_{data}, p_{pos}, p_{tri}$ should satisfy:

- Random positive sampling: $\forall x, y, p_{pos}(x, y) = p_{data}(x)p_{data}(y|X_x)$, where $p_{data}(\cdot|X_x)$ is the conditional pdf of $p_{data}$ on the set of samples $X_x$ similar to $x$ i.e. $p_{data}(\cdot|X_x) = \frac{p_{data}(\cdot)}{p_{data}(X_x)}$.

- Random negative sampling: $p_{tri}(x, y, x^-) = p_{pos}(x, y)p_{data}^-(x^-)$, where $p_{data}^-(x^-) = \frac{p_{data}(x^-)}{\int_{x^-} p_{data}(x^-)dx^-}$

- Class-balanced learning: $p_{data}(X_i) = \frac{1}{n}$, then $\int_{x^-} p_{data}(x^-)dx^- = \frac{n-1}{n}$

where $x^-$ is a negative sample *w.r.t.* $x$ and $n$ is the number of classes.

**Definition 3.** *(Triplet loss)*

$$\mathcal{L}_{triplet}(f, \tau) := \mathbb{E}_{(x,y,x^-) \sim p_{tri}} \left[ (||f(x) - f(y)||_2^2 - ||f(x) - f(x^-)||_2^2 + \tau)_+ \right] \qquad (4)$$

Triplet loss can be rewritten into the form of naive linear loss with a different distribution of triplets. We consider a new distribution:

$$p'_{tri} = \begin{cases} 0, & \text{when } ||f(x) - f(y)||_2^2 - ||f(x) - f(x^-)||_2^2 + \tau < 0, \\ Cp_{tri}, & \text{else,} \end{cases}$$

where $C = 1/\mathbb{E}_{(x,y,x^-) \sim p_{tri}}[1_{\{||f(x)-f(y)||_2^2-||f(x)-f(x^-)||_2^2+\tau \geq 0\}}]$, then

$$\mathcal{L}_{triplet}(f, \tau) = \mathcal{L}_{linear}(f; X, p'_{tri}) = \mathbb{E}_{(x,y,x^-) \sim p'_{tri}}[||f(x) - f(y)||_2^2 - ||f(x) - f(x^-)||_2^2].$$

Apparently the positive part of the triplet loss is minimizing the intra-class distance under a new distribution $p'_{tri}$, which has similar effect as the alignment loss with $p_{tri}$ shown in the experiments (Table 8). Now we focus on the negative part of the triplet loss.

**Theorem 1.** *Denote the probability density function (pdf) of $d^2(x, y) := ||f(x) - f(y)||_2^2$ w.r.t. $y \sim p_{data}(\cdot|X_x)$ by $q(d^2(x, y))$. Then the pdf of $u = ||f(x) - f(y)||_2^2 - ||f(x) - f(x')||_2^2 + \tau$ with fixed $x, x'$ is $q(u - \tau + d^2(x, x'))$, let $S(x, x') = \int_0^\infty q(u - \tau + d^2(x, x'))du \in [0, 1]$, then*

$$-\mathbb{E}_{(x,y,x^-) \sim p'_{tri}}[||f(x) - f(x^-)||_2^2] = -\frac{n}{n-1}\mathbb{E}_{x \sim p_{data}, x' \sim p_{data}}[||f(x) - f(x')||_2^2 S(x, x')]$$

$$+\frac{1}{n-1}\mathbb{E}_{(x,x') \sim p_{pos}}[||f(x) - f(x')||_2^2 S(x, x')]$$

The negative triplet loss consists of two parts, where the first part dominants the second part because $n$ is always large in practice. The first part is actually a weighted unbiased regularization with weight $S(x, x')$. We think $S(x, x')$ may help the unbiased regularization to achieve hyperspherical uniformity. Because the closed form of $q(d^2(x, y))$ is intractable, it's impossible to analyze $S(x, x')$ theoretically without any assumptions. We assume $q(d^2(x, y))$ is exponentially distributed and show the gradient flow of negative triplet loss is asymptotically equal to the gradient of Gaussian hyperspherical energy. Therefore in this case the negative triplet loss can lead to hyperspherical uniformity.

**Proposition 2.** *Assume $q(d^2(x, y)) = \frac{1}{A}e^{-Ad^2(x,y)}$, $S(x, x') = \frac{1}{A}e^{-A(d^2(x,x^-)-\tau)}$ and for the network parameter $\theta$, we have*

$$-\nabla_\theta \mathbb{E}_{(x,y,x^-) \sim p'_{tri}}[||f(x) - f(x^-)||_2^2] = \frac{e^{A\tau}n}{A^2(n-1)}\nabla_\theta E_G(A, X) + O(\frac{1}{n})$$

*the negative triplet loss has asymptotically the same gradient as Gaussian hyperspherical energy.*

Despite the theoretical analysis, we also empirically show that the negative triplet loss achieves hyperspherical uniformity without any assumption on $S(x, x')$. Besides, the negative part of other metric losses are also shown to achieve hyperspherical uniformity in our empirical study. The details is shown in Appendix C.

In summary, the tuple-based metric losses on the unit hypersphere are closely related to intra-class alignment and hyperspherical uniformity. The positive metric losses target at minimizing the intra-class alignment and the negative metric losses try to keep all samples distributed uniformly on the hypersphere.

**Connection to adversarial examples and adversarial training.** The goal of adversarial examples is to fool the neural network by reducing the model performance. Attacking alignment loss, which positions the embedding of similar samples apart, or attacking the uniformity loss, which pulls dissimilar samples together, can definitely destroy the representation learned by DML models. Thus alignment and uniformity loss are suitable objectives for generating adversarial examples.

## 4 Designing New Adversarial DML models Based on Better Understanding of DML Loss

In this section, we introduce our new adversarial DML models: adversarial DML with alignment or uniformity objective (ADML+A or ADML+U). Before we introduce the details of our models, we share our motivation for designing ADML+A/U models by answering the following questions:

***How can adversarial training helps improve DML models?*** One of the most reasonable explanations is that training with adversarial examples brings additional features to neural networks. For example, compared with clean images, adversarial examples make network representations more consistent with salient data features and human perception (Tsipras et al., 2018). Another possible reason is that adversarial examples can be regarded as a data augmentation method, which prevents overfitting of the neural networks. Augmentation techniques which are similar to adversarial training, e.g. using masking out (DeVries & Taylor, 2017) or adding Gaussian noise (Lopes et al., 2019) to regions in images, can help to achieve better performance on image recognizing tasks.

***Why do we need new objectives for adversarial DML models?*** Based on our analysis in Sec. 3, the DML embedding of each image is depend on **all** other positive and negative samples from perspective of alignment and uniformity objective. Therefore if we want to generate the adversarial sample $x'$ for one image $x$, we need to push the adversarial sample away from the similar samples of $x$ (maximize the alignment loss w.r.t. $x'$), or pull the adversarial sample close to the dissimilar samples of $x$ (maximize the uniformity loss w.r.t. $x'$). Currently, the existing adversarial DML models (Duan et al., 2018; Panum et al., 2021) generate adversarial samples by attacking the triplet loss (Definition 3). In this case, only **one** positive and **one** negative samples are used to generating adversarial sample $x'$, which is obviously less powerful than the alignment/uniformity objective (which utilizes more positive or negative samples). Our experimental results in Table 1 also show the adversarial examples generated by alignment/uniformity objectives is more powerful than the triplet objective. Thus it is critical to design new objectives for attacking DML models, which can take advantage of the representation information from more data samples.

**Adversarial training.** We first recall the standard tuple-based DML training setting. Denote the metric loss function by $\mathcal{L}(\cdot; \theta)$, where $\theta$ is the model parameters, our learning objective is:

$$\min_{\theta} \mathbb{E}_{(x, x^+, x^-) \sim p}[\mathcal{L}((x, x^+, x^-); \theta)]$$

In the regular adversarial training framework (Madry et al., 2017), we train networks with perturbed samples from distribution $p^{(adv)}$

$$\min_{\theta} \mathbb{E}_{(x, x^+, x^-) \sim p^{(adv)}}[\mathcal{L}((x, x^+, x^-); \theta)$$

As our goal is to improve the DML performance on clean images by leveraging the regularization power of adversarial examples, we treat adversarial images as additional data augmentations and train networks with a mixture of adversarial examples and clean images. Our learning objective is

$$\min_{\theta}(\mathbb{E}_{(x, x^+, x^-) \sim p}[\mathcal{L}((x, x^+, x^-); \theta)] + \lambda \mathbb{E}_{(x, x^+, x^-) \sim p^{(adv)}}[\mathcal{L}((x, x^+, x^-); \theta)]) \quad (5)$$

where $\lambda$ is the strength of the adversarial training.

Table 1: Performance of DML model evaluated on adversarial samples generated by different objectives. The threatened model is a pretrained Triplet-D model. Adversarial samples are generated from triplet, alignment or uniformity loss. Lower score indicates better quality of adversarial samples.

| Attack objective | CUB200-2011 | | | CARS196 | | | Online-products | | |
|---|---|---|---|---|---|---|---|---|---|
| | R@1 | NMI | mAP@C | R@1 | NMI | mAP@C | R@1 | NMI | mAP@C |
| No attack | 62.40 | 67.21 | 23.56 | 77.59 | 66.64 | 23.83 | 77.53 | 89.98 | 41.12 |
| Triplet | 28.33 | 44.48 | 6.61 | 22.42 | 34.30 | 3.47 | 53.77 | 83.43 | 25.21 |
| Alignment | **8.71** | **23.42** | **0.46** | **10.99** | **17.11** | **0.55** | 8.82 | 80.05 | 1.49 |
| Uniformity | 13.47 | 32.87 | 3.54 | 14.30 | 24.07 | 2.05 | **4.68** | **79.90** | **0.51** |

**Generate adversarial samples.** We use $l_\infty$ PGD-FSGM (Madry et al., 2017) method for generating adversarial samples. We consider DML models require class-balanced batches for training, and propose to generate perturbations by maximizing the intra-batch alignment or uniformity. Given a batch $S$ and a sample $x$, the $l_\infty$ FSGM adversarial sample of $x$ generated by maximizing the alignment objective is given by,

$$\text{ADV}(x) := \Pi_{B_\infty(x,\epsilon)}(x + \alpha \nabla_x \mathcal{L}_{alignment}(f; S)) \tag{6}$$

where $\Pi_{B_\infty(x,\epsilon)}$ is the projecting function on the $l_\infty$ ball centering at $x$ with radius $\epsilon$, and $\alpha$ is the attack strength. Analogously, the adversarial sample generated by maximizing the uniformity objective (Eq. 2 and Eq. 3) is

$$\text{ADV}(x) := \Pi_{B_\infty(x,\epsilon)}(x + \alpha \nabla_x \mathcal{L}_{uniformity}(f; S)) \tag{7}$$

In PGD-FSGM method, we will update the adversarial samples iterative by $x^{(l+1)} = ADV(x^{(l)})$ for $L$ steps. The output perturbed samples $x^{(L)}$ will be used in our adversarial training objective Eq. 5. In ADML-A we use alignment loss (Eq. 1) to generate adversarial samples and in ADML-U we use Gaussian uniformity loss G-HE (Eq. 3). We include the algorithm of ADML-A and ADML-U in the appendix (Alg. 1 and Alg. 2). In experiments, we apply multi-similarity losses as the metric loss $\mathcal{L}$ with attacking alignment and uniformity objective, both models achieve significantly better results on benchmarks (Table 2 and Table 3).

## 5 EXPERIMENTS

In our experiments, we first conduct the empirical studies to verify the theoretical analysis results in Sec. 3 (the results are discussed in Appendix C). After that, we show that alignment and uniformity objectives can help to generate better adversarial examples than the triplet loss, then we compare the natural and robust performance of our adversarial DML model with the state-of-the-art methods. The *natural performance* is the performance of DML models evaluated with clean samples, while the *robust performance* is evaluated with adversarial samples.

### 5.1 EXPERIMENTAL SETUP

**Datasets.** We test our model on four DML benchmarks, CUB200-2011 (Wah et al., 2011), CARS196 (Krause et al., 2013), Online-product (Song et al., 2016), and In-shop (Liu et al., 2016). We follow the previous work (Song et al., 2016) and (Liu et al., 2016) for the train-test split. The statistics of these datasets is introduced in Sec. D.2

**Training Frameworks.** In all experiments, we use the DML framework from (Roth et al., 2020) for training. This framework enables us to train and evaluate DML models under the same settings and ensure fair comparison of the model performance. The backbone network is ResNet50 (He et al., 2016) with ImageNet pretrained (Krizhevsky et al., 2012) and frozen Batch-Normalization layers, the embedding dimension of samples is 128. The initial learning rate is 0.00001 with no scheduling and the batch size is 112. Experiments are performed on a 24GB Nvidia Tesla P40.

**Baseline models.** We compare ADML+A/U with the state-of-the-art DML models. First we select three of the best DML models according to (Roth et al., 2020): margin loss with distance sampling (Wu et al., 2017), multisimilarity loss (Wang et al., 2019) and triplet loss with distance sampling. Besides, we take another two SOTA DML models which were published recently but not included

Table 2: Performance of metric learning models on clustering and retrieval tasks averaged over 5 runs on CUB200-2011 and CARS196. Our Adversarial DML models, ADML+A, and ADML+U, outperform the rest models. The model settings and training parameters are same for all models.

| Models | CUB200-2011 | | | CARS196 | | |
|---|---|---|---|---|---|---|
| | R@1 | NMI | mAP@C | R@1 | NMI | mAP@C |
| ImageNet pretrain | 43.77 | 57.56 | 8.99 | 36.39 | 37.96 | 4.93 |
| Linear | 38.42 | 43.28 | 7.64 | 32.45 | 35.12 | 3.48 |
| Triplet-D | 62.31 ± 0.41 | 67.23 ± 0.34 | 23.29 ± 0.25 | 79.08 ± 0.41 | 66.02 ± 0.33 | 24.02 ± 0.31 |
| Margin | 62.42 ± 0.36 | 67.11 ± 0.49 | 23.54 ± 0.21 | 78.11 ± 0.32 | 66.87 ± 0.35 | 23.94 ± 0.27 |
| Multi-Similarity | 62.73 ± 0.61 | 67.45 ± 0.39 | 22.65 ± 0.34 | 79.94 ± 0.28 | 67.59 ± 0.43 | 24.12 ± 0.25 |
| Proxy-Anchor | 64.16 ± 0.48 | 67.84 ± 0.37 | 23.91 ± 0.32 | 80.13 ± 0.33 | 67.31 ± 0.41 | 23.86 ± 0.26 |
| Cross-Entropy | 61.58 ± 0.31 | 66.67 ± 0.39 | 22.25 ± 0.20 | 78.41 ± 0.39 | 66.35 ± 0.31 | 23.63 ± 0.34 |
| Info-NCE | 61.79 ± 0.51 | 66.91 ± 0.42 | 22.43 ± 0.28 | 77.52 ± 0.37 | 66.75 ± 0.57 | 23.41 ± 0.22 |
| ADML+T | 64.37 ± 0.43 | 68.13 ± 0.49 | 24.05 ± 0.30 | 80.88 ± 0.46 | 66.47 ± 0.51 | 23.91 ± 0.39 |
| ADML+A | **66.02 ± 0.35** | **68.78 ± 0.37** | 24.46 ± 0.23 | 81.95 ± 0.38 | 67.97 ± 0.49 | 24.21 ± 0.28 |
| ADML+U | 65.46 ± 0.40 | 68.60 ± 0.33 | **24.58 ± 0.28** | **82.06 ± 0.36** | **68.21 ± 0.35** | **24.82 ± 0.34** |
| ADML+A+U | 64.24 ± 0.38 | 67.73 ± 0.45 | 23.88 ± 0.26 | 80.95 ± 0.41 | 67.64 ± 0.39 | 23.85 ± 0.30 |

in Roth et al. (2020)'s work: Proxy-Anchor loss (Kim et al., 2020) and Cross-Entropy loss (Boudiaf et al., 2020). Next we apply the Info-NCE loss (Wang & Isola, 2020), which is a contrastive learning objective, as one of the baselines. Finally we compare our models with the only existing adversarial DML model (Duan et al., 2018), ADML+T (triplet objective), in the experiments.

**Evaluation Metrics.** We measure the performance of DML and ADML models with Recall@k (R@k) (Jegou et al., 2010), Normalized Mutual Information (NMI) (Christopher et al., 2008) and Mean Average Precision measured on recall (mAP@k) (Musgrave et al., 2020). The details of these metrics are introduced in Sec. D.3.

## 5.2 COMPARE THE QUALITY OF ADVERSARIAL EXAMPLES WITH DIFFERENT OBJECTIVES

**Settings.** The threatened model is a pretrained DML model with triplet loss and distance sampling (Wu et al., 2017), which is one of the most competitive DML models according to Roth et al. (2020). We consider three different attack objectives, triplet loss (Eq. 4), alignment loss (Eq. 1), and Gaussian hyperspherical uniformity (Eq. 3), for generating adversarial samples. We use $l_\infty$ PGD-FSGM attacks with strength $\epsilon = 0.0314$, $L = 7$ steps and step size $\alpha = 0.007$, we keep this settings for all three attack objectives in our experiments.

**Results.** In Table 1, the adversarial samples generated by alignment or uniformity objectives are significantly stronger than the samples generated by triplet loss. This indicates that adversarial samples from alignment or uniformity contain more features that are not captured by the vanilla DML models. Thus we believe the Adversarial DML model with alignment or uniformity objectives could be more generalized than the vanilla DML models or ADML with triplet loss. Our experimental results in Table 2 and Table 3, which show that ADML+A and ADML+U outperform the baseline models on metric learning tasks, also support our analysis. We also notice that using both A and U in ADML have similar performance as ADML+T, which suggests we should use the attack objective (A or U) separately. In Appendix F we also shows the T-SNE plot of the embedding generated by the vanilla DML and ADML+A, which shows ADML+A can better separate different classes.

## 5.3 ADVERSARIAL DML MODELS IMPROVE NATURAL PERFORMANCE

**Settings.** We train all DML models for 100 epochs. For our adversarial DML models we apply ADML+A and ADML+U. The adversarial training strength $\lambda$ in ADML+A/U is 0.1 for CUB200-2011 and 0.15 for CARS196. For generating adversarial examples we use $l_\infty$ PGD-FSGM attacks with strength $\epsilon = 0.0314$, $L = 7$ steps and step size $\alpha = 0.007$. For Online-products and In-shop we use $\lambda = 0.005$, $\epsilon = 0.01$, $L = 5$, and $\alpha = 0.003$. The ImageNet model is the model only with ImageNet pretrain.

**Results.** In Table 2, our ADML+A and ADML+U models outperform the SOTA metric learning models. ADML+A/U improve the R@1 over 1% on CUB200-2011 and CARS196, and also have considerable improvement on Online-product and In-shop dataset. Besides, the performance of ADML+A/U under the NMI and mAP@C metrics is also comparable or better than the SOTA.

Table 3: Performance of DML models on clustering and retrieval tasks averaged over 5 runs on Online-products and In-shop. Our Adversarial DML models, ADML+A, and ADML+U, outperform the rest models in most cases. The model settings and training parameters are same for all models.

| Models | Online-product | | | In-shop | | |
|---|---|---|---|---|---|---|
| | R@1 | NMI | mAP@C | R@1 | NMI | mAP@C |
| ImageNet pretrain | 48.51 | 84.24 | 17.37 | 21.62 | 76.53 | 4.02 |
| Linear | 20.53 | 81.20 | 5.78 | 16.03 | 75.81 | 2.47 |
| Triplet-D | 77.41 ± 0.19 | **90.04 ± 0.05** | 41.05 ± 0.14 | 87.31 ± 0.18 | 89.76 ± 0.09 | 28.45 ± 0.17 |
| Margin | 77.66 ± 0.14 | 89.93 ± 0.06 | 41.41 ± 0.12 | 87.56 ± 0.15 | **89.93 ± 0.07** | 28.45 ± 0.13 |
| Multi-Similarity | 77.75 ± 0.11 | 90.00 ± 0.04 | 41.39 ± 0.10 | 87.33 ± 0.20 | 89.85 ± 0.12 | 29.61 ± 0.16 |
| Proxy-Anchor | 77.11 ± 0.13 | 89.90 ± 0.05 | 40.98 ± 0.15 | 87.14 ± 0.17 | 89.41 ± 0.05 | 28.11 ± 0.11 |
| Cross-Entropy | 76.92 ± 0.36 | 89.82 ± 0.11 | 41.31 ± 0.42 | 86.75 ± 0.32 | 89.71 ± 0.13 | 28.38 ± 0.51 |
| Info-NCE | 76.21 ± 0.15 | 89.71 ± 0.04 | 39.42 ± 0.09 | 86.24 ± 0.14 | 89.62 ± 0.04 | 27.94 ± 0.17 |
| ADML+T | 77.13 ± 0.11 | 89.59 ± 0.03 | 40.75 ± 0.07 | 87.47 ± 0.12 | 89.65 ± 0.10 | 29.05 ± 0.12 |
| ADML+A | **78.12 ± 0.16** | 89.95 ± 0.04 | **41.56 ± 0.11** | **87.94 ± 0.15** | 89.93 ± 0.05 | **30.12 ± 0.15** |
| ADML+U | 78.01 ± 0.12 | 89.97 ± 0.03 | 41.21 ± 0.12 | 87.86 ± 0.18 | 89.57 ± 0.08 | 29.93 ± 0.16 |
| ADML+A+U | 77.41 ± 0.15 | 89.88 ± 0.07 | 40.91 ± 0.14 | 87.65 ± 0.12 | 89.71 ± 0.09 | 29.72 ± 0.13 |

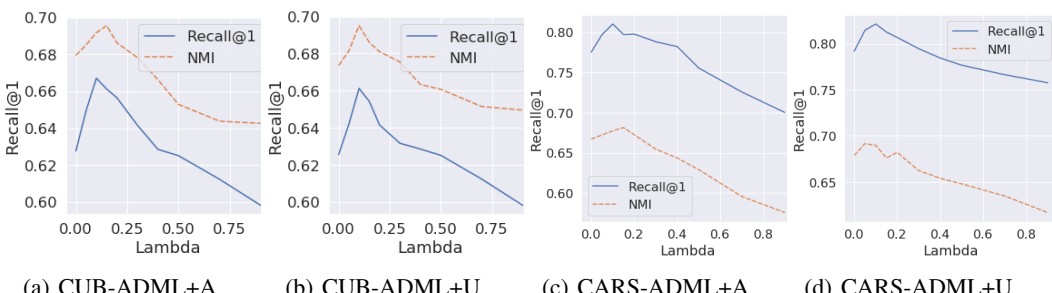

| (a) CUB-ADML+A | (b) CUB-ADML+U | (c) CARS-ADML+A | (d) CARS-ADML+U |
|---|---|---|---|

Figure 1: Performance of ADML+A and ADML+U with different adversarial training strength $\lambda$ on CUB200-2011 and CARS196.

Since all the models are training under the same framework and settings, we can conclude that adversarial training helps to enhance the natuaral performance of DML models.

## 5.4 ROBUSTNESS PERFORMANCE OF ADVERSARIAL DML MODELS

We evaluate the robustness performance of our ADML+A and ADML+U models against attacking the alignment objective, which is the strongest DML attack according to our experiment in Sec. 5.2, on CUB200-2011 and CARS196 datasets. For baseline models we use margin, multi-similarity, and ADML+Triplet models. The settings of alignment attack are the same as the settings in Sec. 5.2, the settings of ADML+A/U and baseline models are the same as the settings in Sec. 5.3. As seen in Table 4, ADML+T has slightly better performance than the vanilla DML models, while ADML+A and ADML+U outperform the baseline models with a large margin under the alignment attacks. Notice ADML+A takes the advantage of adversarial training with alignment loss, it's reasonable that the performance ADML+A is slightly better than ADML+U under alignment attacks.

Table 4: Robustness performance of DML models and ADML models against adversarial samples generated by attacking alignment loss.

| Models | CUB200-2011 | | | CARS196 | | | Online-Products | | |
|---|---|---|---|---|---|---|---|---|---|
| | R@1 | NMI | mAP@C | R@1 | NMI | mAP@C | R@1 | NMI | mAP@C |
| Margin | 8.04 | 24.59 | 0.58 | 8.39 | 17.17 | 0.56 | 8.27 | 80.02 | 1.34 |
| Multi-similarity | 8.90 | 24.13 | 0.51 | 11.95 | 18.32 | 0.55 | 8.93 | 80.06 | 1.45 |
| ADML+T | 11.58 | 25.26 | 0.74 | 25.35 | 21.22 | 1.31 | 10.69 | 80.20 | 1.74 |
| ADML+A | **17.37** | **29.16** | **1.27** | **39.97** | **26.05** | **2.54** | **13.98** | **80.40** | **2.01** |
| ADML+U | 15.10 | 27.89 | 1.06 | 33.09 | 24.48 | 2.30 | 11.32 | 80.29 | 1.85 |

Table 5: Robustness with different adversarial training strength $\lambda$ on CUB200-2011 under the alignment attacks. The metric is Recall@1.

| $\lambda$ | 0 | 0.2 | 0.4 | 0.6 | 0.8 | 1.0 |
|---|---|---|---|---|---|---|
| ADML+A | 8.90 | 19.15 | 22.51 | 24.47 | 25.29 | 25.93 |
| ADML+U | 8.90 | 16.21 | 18.36 | 20.23 | 21.06 | 21.55 |

### 5.5 ABLATION STUDY

**Strength of adversarial training.** In this part, we evaluate the effect of adversarial training strength $\lambda$ on the performance of ADML+A and ADML+U. The backbone is DML with multi-similarity loss. We expect the performance will first increase with $\lambda$ then decrease, because when $\lambda$ is close to 0, the ADML models can hardly learn adversarial features and the improvement is small, when $\lambda$ is large, the adversarial features will dominate the DML models and the performance on clean features will be poor. The model settings are the same as in Sec. 5.3. The experimental results illustrated in Fig. 1 are consistent with our analysis, when increasing the adversarial training strength, the natural performance of ADML+A and ADML+U is first improved then decreased. Table 5 shows the robustness of ADML+A and ADML+U model with adversarial training strength $\lambda$ and alignment attacks. Both models become increasingly robust against alignment attacks.

**ADML on different metric loss.** In this experiment, we apply our ADML approach with triplet, alignment, uniformity, and alignment+uniformity objectives on different metric losses, including triplet, margin, multi-similarity, and info-NCE losses. The metric is Recall@1. The model settings are the same as in Sec. 5.3. From Table 6, we can observe that all ADML methods boost the performance of all DML losses. ADML+A achieves the most significant improvement across all attacks and metric losses.

Table 6: Recall@1 of different metric learning losses with ADML methods on CUB200-2011

| | Vanilla | ADML+T | ADML+A | ADML+U | ADML+A+U |
|---|---|---|---|---|---|
| Triplet | 62.29 | 63.68 | 65.11 | 64.72 | 63.17 |
| Margin | 62.48 | 64.26 | 65.92 | 65.61 | 64.32 |
| Multi-similarity | 62.71 | 64.45 | 66.13 | 65.58 | 64.39 |
| Info-NCE | 61.42 | 62.74 | 64.01 | 63.85 | 62.91 |

**A mixture of alignment and uniformity attacks.** In this experiment, we study the effect of different weight of alignment and uniformity objectives in ADML+A+U. The backbone is DML with multi-similarity loss. The model settings are the same as in Sec. 5.3. Specifically, we assign weight $(1-\beta)$ to the alignment objective and $\beta$ to the uniformity objective. The results are shown in Table 7. When $\beta$ increases, the performance of the ADML model first decreases and then increases, which indicates that attacking alignment and uniformity loss separately can lead to better results.

Table 7: Recall@1 of ADML+$(1-\beta)$A+$\beta$ U on CUB200-2011 with different $\beta$.

| $\beta$ | 0 | 0.2 | 0.4 | 0.5 | 0.6 | 0.8 | 1 |
|---|---|---|---|---|---|---|---|
| ADML+$(1-\beta)$A+$(\beta)$ U | 66.13 | 65.81 | 64.67 | 64.39 | 64.55 | 65.33 | 65.58 |

## 6 CONCLUSION

In this work, we investigated two important properties, intra-class alignment and hyperspherical uniformity, of tuple-based metric losses on unit sphere. According to our theoretical analysis and experimental results, the positive metric losses contribute to the intra-class alignment and the negative metric losses achieve hyperspherical uniformity. Based on our new understanding, we design two novel adversarial DML models, ADML+A and ADML+U, where the perturbations are generated by maximizing the alignment loss or the uniformity loss. Our ADML+A and ADML+U improve both of natural and robust DML performance by enhancing model generalization. Potential future work directions include analyzing other tuple-based metric losses theoretically, designing new metric losses based on the alignment and uniformity, and exploring the trade-off between nature and robustness performance in metric learning.

ETHICS STATEMENT

In this paper, we design new adversarial deep metric learning (DML) models for improving the vanilla DML. Our method can boost the robustness of neural networks again adversarial attacks, which contributes to the trustworthy AI and machine learning.

REPRODUCIBILITY

All datasets, baseline models, general training settings are provided in Sec. 5.1. For specific tasks we also include the detailed settings in the corresponding sections. For example, the detailed model structure and hyperparameter settings for the natural performance of ADML models are written in Sec. 5.3. We will release the code if the paper is accepted.

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

## APPENDIX A   CODE

We use the open source DML training framework `https://github.com/Confusezius/Deep-Metric-Learning-Baselines`. We include the code of our Alg. 1 and Alg. 2 in the supplementary material. All datasets used in this paper are available online. We will release the full code package if our paper is accepted.

## APPENDIX B   THEORETICAL EXPLANATION OF POSITIVE AND NEGATIVE METRIC LOSSES

### B.1   NUMBER OF NEARLY ORTHOGONAL VECTORS IN HIGH DIMENSIONAL SPHERE.

**Lemma 1.** *(Kabatjanskii-Levenstein bound (Kabatjanskii & Levenstein, 1978)) For a hypersphere $\mathcal{S}^{k-1} \in \mathbb{R}^k$, there exist at least $k^M$ vectors with pairwise distance in the range of $\sqrt{2} \pm O(\sqrt{\frac{M \log k}{k}})$.*

Lemma 1 shows the number of nearly orthogonal vectors we can take from a high dimensional sphere. If $k = 128$ and $M = 3$, the amount of nearly orthogonal vectors could be more than one million, which exceeds the volume of benchmark datasets $X$ in DML a lot. Following the former discussion, we know the pairwise distance cannot be larger than $\sqrt{2}$ for all negative pairs. Therefore we think the uniformity regularization with DML benchmarks will lead to an embedding space where the feature vectors are nearly orthogonal. The plots of pairwise distance distribution in Fig. 2 with hyperspherical regularization also support our analysis.

### B.2   NAIVE LINEAR METRIC LOSSES

With the same Assumption in Sec. 3.1, we can prove that the positive part of the linear loss minimizes the intra-class distance and the negative part maximizes an unbiased term, which doesn't achieve hyperspherical uniformity. The theoretical analysis results are summarized in the following theorem.

**Definition 4.** *(Naive linear loss)*

$$\mathcal{L}_{linear}(f; X, p_{tri}) := \mathbb{E}_{(x,y,x^-) \sim p_{tri}} \left[ ||f(x) - f(y)||_2^2 - ||f(x) - f(x^-)||_2^2 \right].$$

**Definition 5.** *(Unbiased regularization)*

$$\mathcal{L}_{unbiased}(f; X, p_{data}) := ||\mathbb{E}_{x \sim p_{data}}[f(x)]||_2^2.$$

*the minimum of this loss is reached when the centroid coincide with the origin.*

**Theorem 2.** *Naive linear loss consists of alignment loss and unbiased loss with a constant multiplier.*

*Positive part:* $\mathbb{E}_{(x,y,x^-) \sim p_{tri}} \left[ ||f(x) - f(y)||_2^2 \right] = \mathcal{L}_{alignment}$

*Negative part:* $- \mathbb{E}_{(x,y,x^-) \sim p_{tri}} \left[ ||f(x) - f(x^-)||_2^2 \right] = \frac{n}{n-1}(2\mathcal{L}_{unbiased} - 2) + \frac{1}{n-1}\mathcal{L}_{alignment}$

*Combining them we have $\mathcal{L}_{linear} = \frac{n}{n-1}(2\mathcal{L}_{unbiased} + \mathcal{L}_{alignment} - 2)$.*

Because the number of classes $n$ is always large and the magnitude of $\mathcal{L}_{unbiased} - 1$ and $\mathcal{L}_{alignment}$ are similar, in negative linear loss the dominant objective is the unbiased regularization. Thus simply maximizing the distance between negative pairs will lead to the unbiased regularization instead of hyperspherical uniformity. In Figure 2(e) and Figure 2(c) of the appendix, we can observe the difference between the models with negative linear loss and uniformity regularization. We also show the interesting connection between the naive linear loss and linear discriminant analysis (LDA) in Sec. E.5.

**Remark.** Naive linear loss doesn't work at all in practice (see the experimental results of Linear model in Table 2 and Table 3). Based on our analysis, we believe it's because linear loss doesn't optimize the hyperspherical uniformity. In the next section, we will introduce our theoretical analysis of triplet loss, which is a simple variant of linear loss. We find triplet loss optimize the hyperspherical uniformity, which could be the reason that triplet loss works well empirically.

## APPENDIX C   EMPIRICAL STUDY OF POSITIVE AND NEGATIVE METRIC LOSSES

In this subsection, we study the effect of tuple-based metric losses on positive or negative pairs empirically, and focus on the four tuple-based losses: naive linear, triplet, margin, and multi-similarity loss. In experiments we train all DML models with either positive metric losses or negative metric losses.

### C.1   DML MODELS WITH POSITIVE METRIC LOSSES

**Settings**. We train 4 DML models with linear, triplet, margin, and multi-similarity losses on the positive sample pairs. The gradient flow of negative pairs is stopped. We train all models for 50 epochs. We compare the average distance between positive/negative/all pairs of embedded samples.

**Results.** In Table 8, the DML models trained with only positive metric losses have average pairwise distance close to 0. Therefore, minimizing the intra-class alignment will lead to a model which maps all samples to the same feature vector.

Table 8: Comparison of DML models trained with the positive metric losses on CUB200-2011 and CARS196. We calculate the average distance (Avgdist), average distance of positive pairs (Avgdist-Pos), and average distance of negative pairs (AvgdistNeg) with the embedded samples.

|  | CUB200-2011 | | | CARS196 | | |
|---|---|---|---|---|---|---|
|  | AvgdistPos | AvgdistNeg | Avgdist | AvgdistPos | AvgdistNeg | Avgdist |
| Linear | 2.010e-3 | 3.190e-3 | 3.178e-3 | 1.497e-3 | 3.162e-3 | 3.183e-3 |
| Triplet | 1.934e-3 | 3.187e-3 | 3.174e-3 | 1.476e-3 | 3.163e-3 | 3.184e-3 |
| Margin | 7.565e-2 | 8.200e-2 | 8.192e-2 | 3.195e-2 | 3.468e-2 | 3.465e-2 |
| MS | 2.558e-3 | 3.324e-3 | 3.316e-3 | 2.242e-3 | 3.264e-3 | 3.277e-3 |

### C.2   COMPARE NEGATIVE METRIC LOSSES WITH UNIFORMITY REGULARIZATION

**Settings.** We compare 8 different models in this experiment, including a) the original model with only ImageNet pretrain; b) four models trained with linear, triplet, margin, and multi-similarity losses on negative pairs; c) three models trained with HE(s=0), HE(s=1) (Eq. 2) and G-HE(s=1) (Eq. 3) regularization, those regularization functions are introduced in Appendix C. We train all models with 50 epochs. For the models with negative metric losses, the gradient flow of positive pairs is stopped.

**Evaluation Metrics.** We use the regularization score on HE(s=0) (Eq. 2) and G-HE(s=1) (Eq. 3) to measure the uniformity of embedded samples on hypersphere. Smaller score indicates better uniformity. We also check the average of pairwise distance of different models, we expect it to be close to $\sqrt{2}$ for good hyperspherical uniformity.

**Results.** In experiments we compare the regularization strength of negative metric losses with HE and G-HE. Fig. 2 and Fig. 3 illustrates the pairwise distance of the test samples on CUB200-2011 and CARS196 dataset. The DML models with negative metric losses have similar pairwise distance distributions as the uniformity regularization methods, and the only exception is the negative naive linear loss, which will not lead to the hyperspherical uniformity based on our analysis in Sec. B.2. We also include the distance distribution before training (Figure 2(a)) as a reference. Besides, we also compare those models under the hyperspherical uniformity metrics. In Table 9, the negative metric losses achieve comparable results with the uniformity regularization methods. The G-HE(s=1) outperforms the rest models.

Table 9: Comparison of DML models trained with the negative metric losses and uniformity regularization on CUB200-2011 and CARS196. All negative metric losses except Linear achieve comparable performance to uniformity regularization. The details of models and training settings are in Sec. C.2.

| | CUB200-2011 | | | CARS196 | | |
|---|---|---|---|---|---|---|
| | Avgdist | HE(s=0) | G-HE(s=1) | Avgdist | HE(s=0) | G-HE(s=1) |
| ImageNet | 0.7220 | 0.3368 | 0.5939 | 0.6557 | 0.4326 | 0.6497 |
| Linear | 1.2398 | -0.1670 | 0.2535 | 1.3194 | -0.1686 | 0.2449 |
| Triplet | 1.3877 | -0.3254 | 0.1496 | 1.4039 | -0.3376 | 0.1420 |
| Margin | 1.3929 | -0.3297 | 0.1465 | 1.4016 | -0.3363 | 0.1426 |
| MS | 1.3825 | -0.3221 | 0.1509 | 1.3985 | -0.3338 | 0.1441 |
| HE regularization (s=0) | 1.4001 | -0.3347 | 0.1438 | 1.4064 | -0.3395 | 0.1411 |
| HE regularization (s=1) | 1.4009 | -0.3355 | 0.1432 | 1.4068 | -0.3398 | 0.1407 |
| G-HE regularization (s=1) | **1.4028** | **-0.3369** | **0.1424** | **1.4077** | **-0.3406** | **0.1402** |

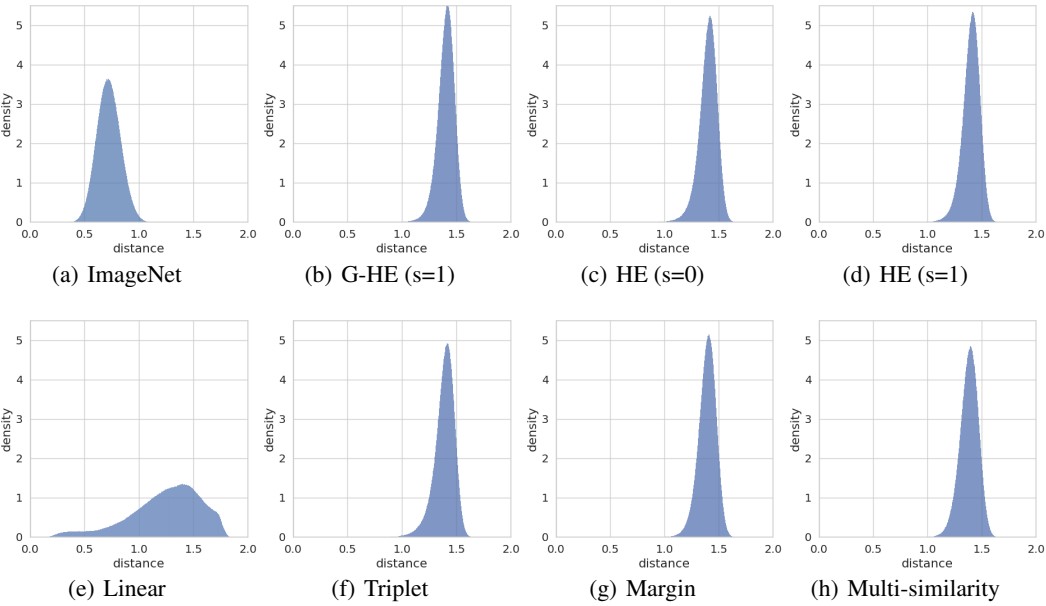

Figure 2: Illustration of pairwise distance distributions of the embedded CUB200-2011 samples generated by DML models trained with negative metric losses or uniformity regularization. The details of models and training settings are in Sec. C.2.

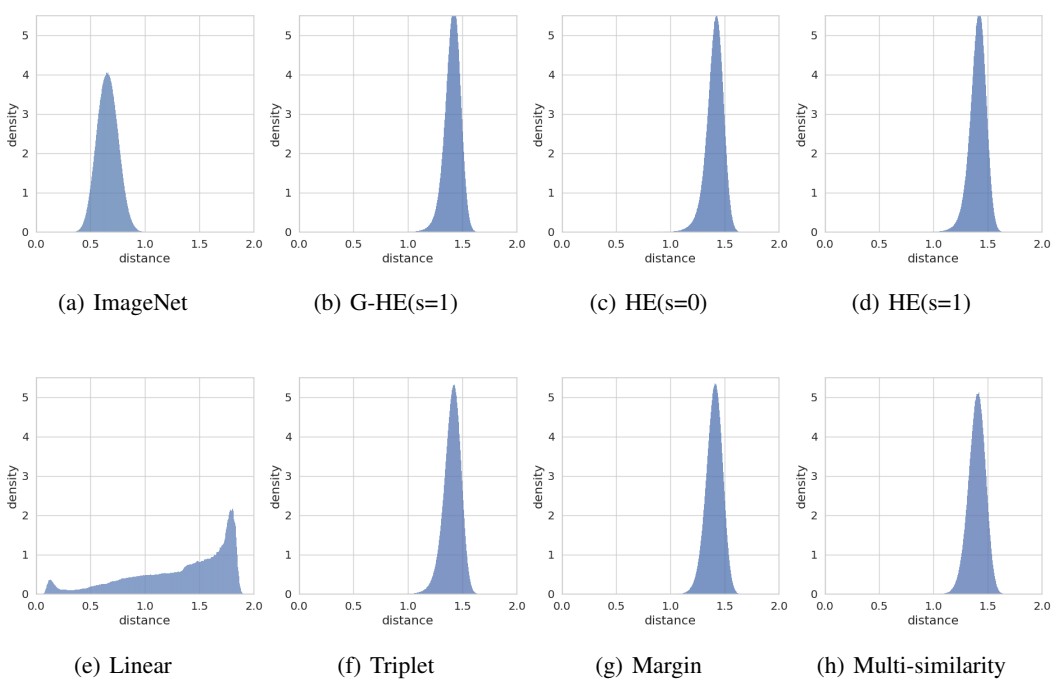

Figure 3: Illustration of pairwise distance distributions of the embedded CARS196 samples with negative metric losses or uniformity regularization. The details of models and training settings are in Sec. 5.3.

## APPENDIX D    ADML ALGORITHMS AND EXPERIMENTAL SETTINGS

### D.1    ALGORITHM OF ADML+A AND ADML+U

---

**Algorithm 1** ADML+A

---

**Input:** training set $X$; number of epochs $N$; original classifier $f$; weight of adversarial training $\lambda$;
   PGD attack step $L$; PGD attack strength $\epsilon$
Initialize class balanced sampler $S$;
 **for** $i \in epochs$ **do**
   **for** $S \in mini\text{-}batch\{S_1, ..., S_n\}$ **do**
      $S^{(0)} = S$;
      Generate adversarial samples of $S$ with PGD-FSGM attack
      **for** $t \in 0 : L - 1$ **do**
         $S^{(t+1)} = \Pi_{B_\infty(S^{(0)}, \epsilon)}(S^{(t)} + \alpha \nabla_{S^{(t)}} \mathcal{L}_{alignment}(f, S^{(t)}))$;
      **end**
      $S_{adv} = S^{(L)}$;
      Calculate objective function
      $\mathcal{L}_{total} = \mathcal{L}(f, S) + \lambda \mathcal{L}(f, S_{adv})$;
      $\mathcal{L}$ can be an arbitrary metric loss, in our paper we use margin and multisimilarity loss
      Update network parameters of $f$ with $\mathcal{L}_{total}$;

   **end**
 **end**

---

---

**Algorithm 2** ADML+U

---

**Input:** training set $X$; number of epochs $N$; original classifier $f$; weight of adversarial training $\lambda$;
        PGD attack step $L$; PGD attack strength $\epsilon$
Initialize class balanced sampler $S$;
 **for** $i \in epochs$ **do**
    **for** $S \in mini\text{-}batch\{S_1, ..., S_n\}$ **do**
       $S^{(0)} = S$;
       Generate adversarial samples of $S$ with PGD-FSGM attack
       **for** $t \in 0 : L - 1$ **do**
          $S^{(t+1)} = \Pi_{B_\infty(S^{(0)},\epsilon)}(S^{(t)} + \alpha\nabla_{S^{(t)}}\mathcal{L}_{uniformity}(f, S^{(t)}))$;
       **end**
       $S_{adv} = S^{(L)}$;
       Calculate objective function
       $\mathcal{L}_{total} = \mathcal{L}(f, S) + \lambda\mathcal{L}(f, S_{adv})$;
       $\mathcal{L}$ can be an arbitrary metric loss, in our paper we use margin and multisimilarity loss
       Update network parameters of $f$ with $\mathcal{L}_{total}$;
    **end**
 **end**

---

## D.2 DATASET DETAILS

- **CUB200-2011** contains 200 species of birds and 11,788 images (Wah et al., 2011). We use the first 100 species as training set and the rest as test set.

- **CARS196** has 196 models of cars and 16,185 images. (Krause et al., 2013). We use the first 98 models as training set and the rest as test set.

- **Online-product** includes 22,634 classes of products and 120,053 images (Song et al., 2016). We use the first 11,318 classes as training set and the rest as test set.

- **In-shop** contains 7982 classes of clothing and 54,624 images (Liu et al., 2016). We use the first 3,997 classes as training set and the rest as test set. The test set is further partitioned into a query set with 14,218 images of 3,985 classes and a gallery set with 12,612 images of 3,985 classes.

## D.3 EVALUATION METRICS

We use the following metrics to evaluate the DML models with retrieval and clustering downstream tasks.

**Recall@k.** For the retrieval task we apply the Recall@k (R@k) metric (Jegou et al., 2010). For a test set $M := \{(x_1, y_1), \cdots, (x_n, y_n)\}$, the indices of the first $k$ nearest neighborhood of a sample $x_i$ is given by $S_k(x_i) := \arg\max_{|S|=k} \sum_{j \in S, j \neq i} ||f(x_i) - f(x_j)||_2$, and

$$\text{R@k} := \frac{1}{n} \sum_{i=1}^{n} 1_{\{\exists j \in S_k(x_i), y_j = y_i\}}.$$

**NMI.** We use Normalized Mutual Information (NMI) (Christopher et al., 2008) to measure the quality of the clustering task. We use K-means to generate the clusters of the embedded samples, then we calculate the label assignment $\Gamma = \{\gamma_1, ..., \gamma_n\}$ from clustering. Denote the ground truth labels by $\Omega = \{y_1, ..., y_n\}$, the NMI is computed as

$$\text{NMI}(\Omega, \Gamma) = I(\Omega, \Gamma)/[2(H(\Omega) + H(\Gamma))],$$

where $I(\cdot, \cdot)$ is the mutual information function and $H(\cdot)$ is the entropy function.

**mAP@C.** According to (Musgrave et al., 2020), we also include mean average precision measured on recall (mAP@k) as an additional metric. We first compute the recalled samples, which are determined by the k nearest neighbour ranking. Then compute the mAP-score follows the standard mAP procedure. mAP@C is the mean over the class-wise average precision@$k_c$, where $k_c$ is the

number of samples in class $c$, which means we only recall $k_c$ nearest neighbour. Following the notation of Recall@k, the value of mAP@C is given by

$$mAP@C := \frac{1}{n} \sum_{c \in C} \sum_{y_q = c} \frac{|\{x_i \in S_{k_c}(x_q)|y_i = y_q\}|}{k_c}$$

.

## APPENDIX E    MISSING PROOFS

### E.1    PROOF OF PROPOSITION 1

Denote the support set of the distribution of each class by $S_1, \ldots, S_n$, according to Definition 1, the minimum of alignment loss is reached when the encoder $f^*$ maps all samples in one class to the same feature vector i.e. $\forall i, f^*(S_i) = \{v_i\}$. For arbitrary $i, j$, because $\cup_{k=1}^n S_k$ is connected and each $S_k$ is closed, we can select a set sequence $S_{k_0}, \ldots, S_{k_m}$ such that $S_{k_0} = S_i, S_{k_m} = S_j$ and $S_{k_l} \cap S_{k_{l+1}} \neq \emptyset, l \in [m-1]$. By $S_{k_l} \cap S_{k_{l+1}} \neq \emptyset$ we have $v_{k_l} = v_{k_{l+1}}$ for all $l \in [m-1]$, thus $\forall i, j, f^*(S_i) = f^*(S_j)$. So all samples are projected to the same feature vector.

### E.2    PROOF OF THEOREM 2

*Proof.* Recall that the naive linear loss is given by

$$\mathcal{L}_{linear}(f; X, p_{tri}) := \mathbb{E}_{(x,y,x^-) \sim p_{tri}}[||f(x) - f(y)||_2^2 - ||f(x) - f(x^-)||_2^2]$$

Consider the positive part

$$\mathbb{E}_{(x,y,x^-) \sim p_{tri}}[||f(x) - f(y)||_2^2] = \mathbb{E}_{(x,y) \sim p_{pos}}[||f(x) - f(y)||_2^2] = \mathcal{L}_{alignment} \quad (8)$$

Consider the negative part

$$-\mathbb{E}_{(x,y,x^-) \sim p_{tri}}[||f(x) - f(x^-)||_2^2] = 2\mathbb{E}_{(x,y,x^-) \sim p_{tri}}[f(x)^T f(x^-)] - 2$$

and

$$\mathbb{E}_{(x,y,x^-) \sim p_{tri}}[f(x)^T f(x^-)]$$

$$= \mathbb{E}_{(x,y) \sim p_{pos}}[f(x)^T \mathbb{E}_{x^- \sim p_{data}^-}[f(x^-)]]$$

$$= \mathbb{E}_{x \sim p_{data}}[f(x)^T \frac{1}{\int_{x^-} p_{data}(x^-)dx^-} (\mathbb{E}_{x' \sim p_{data}}[f(x')] - \mathbb{E}_{x' \sim p_{data}}[f(x')1_{x' \in X_x}])]$$

$$= \frac{n}{n-1}(\mathbb{E}_{x \sim p_{data}}[f(x)^T \mathbb{E}_{x' \sim p_{data}}[f(x')]] - \mathbb{E}_{x \sim p_{data}}[f(x)^T \mathbb{E}_{x' \sim p_{data}}[f(x')1_{x' \in X_x}]])$$

$$= \frac{n}{n-1}(\mathbb{E}_{x \sim p_{data}}[f(x)^T \mathbb{E}_{x' \sim p_{data}}[f(x')]] - \mathbb{E}_{x \sim p_{data}}[f(x)^T p_{data}(X_x)\mathbb{E}_{x' \sim p_{data}(\cdot|X_x)}[f(x')]])$$

$$= \frac{n}{n-1}\mathbb{E}_{x \sim p_{data}}[f(x)]^T \mathbb{E}_{x \sim p_{data}}[f(x)] - \frac{1}{n-1}\mathbb{E}_{(x,y) \sim p_{pos}}[f(x)^T f(y)]$$

$$= \frac{n}{n-1}\mathbb{E}_{x \sim p_{data}}[f(x)]^T \mathbb{E}_{x \sim p_{data}}[f(x)] - \frac{1}{n-1}\mathbb{E}_{(x,y) \sim p_{pos}}[f(x)^T f(y)]$$

$$= \frac{n}{n-1}\mathcal{L}_{unbiased} + \frac{1}{2(n-1)}(\mathcal{L}_{alignment} - 2)$$

$$(9)$$

where we denote the class of sample $x$ by $X_x$.

Combining Eq. 8 and Eq. 9 together, we have

$$\mathcal{L}_{linear}(f) = \frac{2n}{n-1}\mathcal{L}_{unbiased} + \frac{1}{n-1}(\mathcal{L}_{alignment} - 2) - 2 + \mathcal{L}_{alignment}$$

$$= \frac{n}{n-1}(2\mathcal{L}_{unbiased} + \mathcal{L}_{alignment} - 2)$$

$$(10)$$

$\square$

### E.3 PROOF OF THEOREM 2

*Proof.* The triplet loss is

$$
\begin{aligned}
\mathcal{L}_{triplet}(f,\tau) = \mathcal{L}_{linear}(f;X,p'_{tri}) &= \mathbb{E}_{(x,y,x^-)\sim p'_{tri}}[\||f(x)-f(y)\||_2^2 - \||f(x)-f(x^-)\||_2^2] \\
&= \mathbb{E}_{(x,y,x^-)\sim p_{tri}}[(\||f(x)-f(y)\||_2^2 - \||f(x)-f(x^-)\||_2^2)1_{\{\||f(x)-f(y)\||_2^2-\||f(x)-f(x^-)\||_2^2+\tau\geq 0\}}] \\
&= \mathbb{E}_{(x,y,x^-)\sim p_{tri}}[\||f(x)-f(y)\||_2^2 1_{\{\||f(x)-f(y)\||_2^2-\||f(x)-f(x^-)\||_2^2+\tau\geq 0\}}] \\
&\quad - \mathbb{E}_{(x,y,x^-)\sim p_{tri}}[\||f(x)-f(x^-)\||_2^2 1_{\{\||f(x)-f(y)\||_2^2-\||f(x)-f(x^-)\||_2^2+\tau\geq 0\}}]
\end{aligned}
$$

Consider the **negative part**,

$$
\begin{aligned}
&- \mathbb{E}_{(x,y,x^-)\sim p_{tri}}[\||f(x)-f(x^-)\||_2^2 1_{\{\||f(x)-f(y)\||_2^2-\||f(x)-f(x^-)\||_2^2+\tau\geq 0\}}] \\
=& - \mathbb{E}_{x\sim p_{data},x^-\sim p_{data}^-}[\||f(x)-f(x^-)\||_2^2 \mathbb{E}_{y\sim p_{data}(\cdot|X_x)}[1_{\{\||f(x)-f(y)\||_2^2-\||f(x)-f(x^-)\||_2^2+\tau\geq 0\}}]] \\
=& - \frac{n}{n-1}\mathbb{E}_{x\sim p_{data},x'\sim p_{data}}[\||f(x)-f(x')\||_2^2 \mathbb{E}_{y\sim p_{data}(\cdot|X_x)}[1_{\{\||f(x)-f(y)\||_2^2-\||f(x)-f(x')\||_2^2+\tau\geq 0\}}]] \\
&+ \frac{n}{n-1}\mathbb{E}_{x\sim p_{data},x'\sim p_{data}}[\||f(x)-f(x')\||_2^2 1_{x'\in X_x}\mathbb{E}_{y\sim p_{data}(\cdot|X_x)}[1_{\{\||f(x)-f(y)\||_2^2-\||f(x)-f(x')\||_2^2+\tau\geq 0\}}]] \\
=& - \frac{n}{n-1}\mathbb{E}_{x\sim p_{data},x'\sim p_{data}}[\||f(x)-f(x')\||_2^2 \mathbb{E}_{y\sim p_{data}(\cdot|X_x)}[1_{\{\||f(x)-f(y)\||_2^2-\||f(x)-f(x')\||_2^2+\tau\geq 0\}}]] \\
&+ \frac{1}{n-1}\mathbb{E}_{(x,x')\sim p_{pos}}[\||f(x)-f(x')\||_2^2 \mathbb{E}_{y\sim p_{data}(\cdot|X_x)}[1_{\{\||f(x)-f(y)\||_2^2-\||f(x)-f(x')\||_2^2+\tau\geq 0\}}]] \\
=& - \frac{n}{n-1}\mathbb{E}_{x\sim p_{data},x'\sim p_{data}}[\||f(x)-f(x')\||_2^2 S(x,x')] \\
&+ \frac{1}{n-1}\mathbb{E}_{(x,x')\sim p_{pos}}[\||f(x)-f(x')\||_2^2 S(x,x')]
\end{aligned}
$$

where $X_x$ is the set of samples have the same label as $x$. The last equation is based on

$$
\begin{aligned}
S(x,x') &= \int_0^\infty q(u+d^2(x,x')-\tau) = \mathbb{E}_{q(d^2(x,y))}[1_{\{u\geq 0\}}] = \mathbb{E}_{y\sim p_{data}(\cdot|X_x)}[1_{\{u\geq 0\}}] \\
&= \mathbb{E}_{y\sim p_{data}(\cdot|X_x)}[1_{\{\||f(x)-f(y)\||_2^2-\||f(x)-f(x')\||_2^2+\tau\geq 0\}}]
\end{aligned}
$$

$\square$

### E.4 PROOF OF PROPOSITION 2

*Proof.* By $q(d^2(x,y)) = \frac{1}{A}e^{-Ad^2(x,y)}$, the pdf of $u = d^2(x,y) - d^2(x,x') + \tau$ is $\frac{1}{A}e^{-A(u+d^2(x,x')-\tau)}$, then

$$
S(x,x') = \frac{1}{A}\int_0^\infty e^{-A(u+d^2(x,x')-\tau)}du = \frac{1}{A}e^{-A(d^2(x,x')-\tau)}
$$

Consider the gradient of the negative triplet loss $\mathbb{E}_{(x,y,x^-)\sim p'_{tri}}[\||f(x)-f(x^-)\||_2^2]$, during training we first sampling from $p'_{tri}$ then calculate the gradient. In this case the actual gradient flow is given by

$$
-\mathbb{E}_{(x,y,x^-)\sim p'_{tri}}[\nabla_\theta \||f(x)-f(x^-)\||_2^2]
$$

Analogous with the discussion in Sec. E.3, we have

$$
\begin{aligned}
&- \mathbb{E}_{(x,y,x^-)\sim p'_{tri}}[\nabla_\theta \|f(x) - f(x^-)\|_2^2]\\
=& - \frac{n}{n-1}\mathbb{E}_{x\sim p_{data},x'\sim p_{data}}[S(x,x')\nabla_\theta\|f(x)-f(x')\|_2^2]\\
&+ \frac{1}{n-1}\mathbb{E}_{(x,x')\sim p_{pos}}[S(x,x')\nabla_\theta\|f(x)-f(x')\|_2^2]\\
=& - \frac{e^{A\tau}n}{A(n-1)}\mathbb{E}_{x\sim p_{data},x'\sim p_{data}}[e^{-Ad^2(x,x')}\nabla_{d(x,x')}d^2(x,x')\frac{\partial d(x,x')}{\partial\theta}] + O(\frac{1}{n})\\
=& - \frac{e^{A\tau}n}{A(n-1)}\mathbb{E}_{x\sim p_{data},x'\sim p_{data}}[e^{-Ad^2(x,x')}2d(x,x')\frac{\partial d(x,x')}{\partial\theta}] + O(\frac{1}{n})\\
=& \frac{e^{A\tau}n}{A^2(n-1)}\mathbb{E}_{x\sim p_{data},x'\sim p_{data}}[\nabla_{d(x,x')}(e^{-Ad^2(x,x')})\frac{\partial d(x,x')}{\partial\theta}] + O(\frac{1}{n})\\
=& \frac{e^{A\tau}n}{A^2(n-1)}\nabla_\theta\mathbb{E}_{x\sim p_{data},x'\sim p_{data}}[e^{-Ad^2(x,x')}] + O(\frac{1}{n})\\
=& \frac{e^{A\tau}n}{A^2(n-1)}\nabla_\theta E_G(A,X) + O(\frac{1}{n})
\end{aligned}
$$

$\square$

### E.5 CONNECTION BETWEEN NAIVE LINEAR LOSS AND LDA

The intuition of multiple linear discriminant analysis (LDA) is to maximize the inter-class variance while minimizing the intra-class variance. In this section we will show linear metric loss have a similar effect.

**Definition 6.** *(Total variation) For a random vector $x$, the total variation is*

$$
TV(x) := tr(\mathbb{E}[(x - \mathbb{E}[x])(x - \mathbb{E}[x])^T]) = \mathbb{E}[(x-\mathbb{E}[x])^T(x-\mathbb{E}[x])] = \mathbb{E}[x^Tx] - \mathbb{E}[x]^T\mathbb{E}[x] \tag{11}
$$

**Definition 7.** *(Centroid) Define the centroid of samples in an arbitrary set $Y$ by*

$$
c_Y := \mathbb{E}_{y\sim p_{data}(\cdot|Y)}[f(y)] = \frac{1}{p_{data}(Y)}\mathbb{E}_{y\sim p_{data}}[f(y)1_{y\in Y}] \tag{12}
$$

**Proposition 3.** *(Intra-class total variation) The within-class variation*

$$
TV_{intra}(X) := \mathbb{E}_{x\sim p_{data}}[(f(x) - c_{X_x})^T(f(x) - c_{X_x})]
$$

*is proportional to alignment loss*

$$
\mathcal{L}_{aligned}(f) = 2TV_{intra}(X) \tag{13}
$$

*Proof.*

$$
\begin{aligned}
TV_{intra}(X) &= \mathbb{E}_{x\sim p_{data}}[(f(x) - c_{X_x})^T(f(x) - c_{X_x})]\\
&= \sum_{i=1}^n p_{data}(X_i)(\mathbb{E}_{x\sim p_{data}(\cdot|X_i)}[f(x)^Tf(x)] - c_{X_i}^T c_{X_i})\\
&= \frac{1}{n}\sum_{i=1}^n(1 - n\mathbb{E}_{x\sim p_{data},x'\sim p_{data}(\cdot|X_x)}[f(x)^Tf(x')1_{x\in X_i}])\\
&= 1 - \mathbb{E}_{x\sim p_{data},x'\sim p_{data}(\cdot|X_x)}[f(x)^Tf(x')\sum_{i=1}^n 1_{x\in X_i}]\\
&= 1 - \mathbb{E}_{(x,y)\sim p_{pos}}[f(x)^Tf(y)]\\
&= \frac{1}{2}\mathcal{L}_{aligned}(f)
\end{aligned} \tag{14}
$$

$\square$

**Proposition 4.** *(Inter-class total variation) The inter-class total variation*

$$TV_{inter}(X) := \mathbb{E}_{x \sim p_{data}}[(c_{X_x} - c)^T(c_{X_x} - c)],$$

*where $c$ is the centroid of all samples, is proportional to triplet loss*

$$\mathcal{L}_{linear}(f) = -\frac{2n}{n-1}TV_{inter}(X) \tag{15}$$

*Proof.*

$$TV_{inter}(X) = \mathbb{E}_{x \sim p_{data}}[c_{X_x}^T c_{X_x}] - c^T c \tag{16}$$

Firstly,

$$c = \frac{1}{p_{data}(X)}\mathbb{E}_{x \sim p_{data}}[f(x)1_{x \in X}] = \mathbb{E}_{x \sim p_{data}}[f(x)]$$

Hence $c^T c = \mathcal{L}_{unbiased}(f)$.

Next,

$$\begin{aligned}
\mathbb{E}_{x \sim p_{data}}[c_{X_x}^T c_{X_x}] &= \sum_{i=1}^{n} p_{data}(X_i)\mathbb{E}_{x \sim p_{data}(\cdot|X_i)}[c_{X_i}^T c_{X_i}] \\
&= \sum_{i=1}^{n} p_{data}(X_i)c_{X_i}^T c_{X_i} \\
&= \sum_{i=1}^{n} \mathbb{E}_{x \sim p_{data}, y \sim p_{data}(\cdot|X_x)}[f(x)^T f(y)1_{x \in X_i}] \\
&= \mathbb{E}_{x \sim p_{data}, y \sim p_{data}(\cdot|X_x)}[f(x)^T f(y) \sum_{i=1}^{n} 1_{x \in X_i}] \\
&= \mathbb{E}_{x \sim p_{data}, y \sim p_{data}(\cdot|X_x)}[f(x)^T f(y)] \\
&= \mathbb{E}_{(x,y) \sim p_{pos}}[f(x)^T f(y)]
\end{aligned}$$

Thus $\mathbb{E}_{x \sim p_{data}}[c_{X_x}^T c_{X_x}] = 1 - \frac{1}{2}\mathcal{L}_{aligned}(f)$, and $TV_{inter}(X) = 1 - \frac{1}{2}\mathcal{L}_{aligned}(f) - \mathcal{L}_{unbiased}(f) = -\frac{n-1}{2n}\mathcal{L}_{linear}(f)$ □

**Proposition 5.** *(Total variation of the dataset) The total variation of dataset $X$*

$$TV_{total}(X) := \mathbb{E}_{x \sim p_{data}}[(x - c)^T(x - c)],$$

*where $c$ is the centroid of all samples, is proportional to the unbiased loss*

$$\mathcal{L}_{unbiased}(f) = 1 - TV_{total}(X) \tag{17}$$

*Proof.*

$$\begin{aligned}
TV_{total}(X) &= \mathbb{E}_{x \sim p_{data}}[f(x)^T f(x)] - \mathbb{E}_{x \sim p_{data}}[f(x)]^T \mathbb{E}_{x \sim p_{data}}[f(x)] \\
&= 1 - \mathcal{L}_{unbiased}(f)
\end{aligned} \tag{18}$$

□

We can also check if $TV_{total}(X) = TV_{within}(X) + TV_{between}(X)$ holds to validate the proofs above.

## APPENDIX F  T-SNE EVALUATION OF ADML+A AND VANILLA DML.

In experiments, we visualize the embedding of the first 10 classes of CUB200-2011 generated by a vanilla DML model (with multi-similarity loss) and an ADML+A model (with multi-similarity loss). The Recall@1 of the DML model and the ADML+A model are 62.71 and 66.13, respectively. Fig. 4 (a) plots the T-SNE result for vanilla DML and Fig. 4 (b) plots the T-SNE result for ADML+A. Comparing two figures, we can see that ADML+A has better separation on the (red, green, orange) samples.

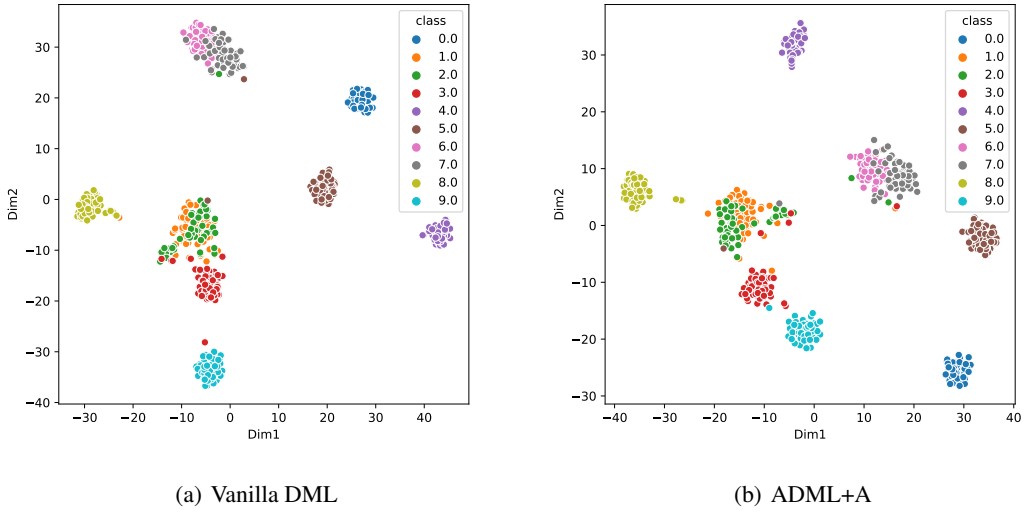

(a) Vanilla DML

(b) ADML+A

Figure 4: T-SNE visualization of embedding generated by a vanilla DML and an ADML+A model on CUB200-2011.

