# OpenReview forum: "Understanding Metric Learning on Unit Hypersphere and Generating Better Examples for Adversarial Training"
_ICLR.cc/2022/Conference — ICLR 2022 Submitted_

### Official Review · Reviewer_MuXo · 2021-11-01

**Correctness:** 4
**Technical Novelty And Significance:** 3
**Empirical Novelty And Significance:** 3
**Recommendation:** 8
**Confidence:** 4

**Main Review:**

Strengths:
- A thorough proof of the link between alignment/uniformity on the sphere is provided for the case of the triplet loss;
- The three variants of the proposed ADML provide a consist performance boost across various benchmarks;
- The relevance of attacking the alignment/uniformity terms is justified empirically (Table 1);
- The effect the hyperparameter $\lambda$ introduces by their training scheme is studied across multiple datasets.

Weaknesses:
- As stated in this paper, separating the triplet loss into the intra-class alignment and the hypersphere uniformity terms is intractable. Therefore it is unclear whether the performance gain is a result of the different loss choice or the robustness to intra-class alignment and the hypersphere uniformity perturbations. In other words:
    - Is the setting ADML+U+A, where both terms are adversarially attack, comparable to ADML+T? If that's the case then this would an argument in favor of attacking each one of the two terms separately. However, if that's not the case, then performance gain might be attributed to the InfoNCE loss being a better attack target;
    - The combination of the intra-class alignment and the hypersphere uniformity terms used to generate adversarial examples is the InfoNCE loss. Would training the ADML model using the InfoNCE loss result in comparable results?
I'm aware the gradients of the intra-class alignment and the hypersphere uniformity terms are only used to generate adversarial samples which are then used to train the model with the triplet loss.

- Table 4 shows that the three ADML variants result in more robust models. However one needs to state that ADML+A has an unfair advantage compared to the rest of the models evaluated since it was explicitly trained for this scenario.

**Summary Of The Paper:**

This work introduces novel adversarial deep metric learning algorithm based on the fact that contrastive losses promote intra-class alignment and uniformity on the unit sphere. The authors provide a theoretical proof that this is also true for the triplet loss under mild assumptions.
The adversarial examples used to regularize the training are generated by targeting the alignment and uniformity terms separately. This regularization strategy results in a significant robustness/performance boost on deep metric learning benchmarks.

**Summary Of The Review:**

This work extends known properties of contrastive losses to the triplet loss. Namely, the loss can be rewritten as the sum of two terms: a intra-class alignment term and a hypersphere uniformity term. Based on this, a novel regularization term was devised for training deep metric learning models based on adversarial attacks on each one of the two terms. The three proposed variants of the ADML model acheive convincing performance gains across different image retrieval benchmarks and are more robust to subsequent adversarial attacks. However, it is unclear whether gain in performance is due to adversarially perturbing the intra-class alignment and the hypersphere uniformity terms separately or due to InfoNCE having more informative gradients compared to the triplet loss.

---

> ### Author Response · Authors · 2021-11-17
> **Author Response to Reviewer MuXo**
>
> Thanks for the reviewer's valuable comments.
>
> *1. ADML+U+A*
>
> --
> This is a nice suggestion. We add experiments with ADML+A+U in our paper (we update Table 2 and Table 3 with ADML+A+U in the revision of our paper). It has worse performance than ADML+A or ADML+U, and similar performance as the ADML+T. Thus, using the separated objective (A or U) can lead to better performance.
>
> Table: ADML+A+U on CUB200-2011 and CARS196
>
> |           |      CUB200-2011      |    |      |        CARS196        |       |    |
> |-----------|:---------------------:|:---------------------:|:---------------------:|:---------------------:|:---------------------:|:---------------------:|
> | Approachs |  R@1    |     NMI       |  mAP@C   |   R@1   | NMI  | mAP@C   |
> |   ADML+T  | 64.37 ± 0.43     | 68.13 ± 0.49     |      24.05 ± 0.30|      80.88 ± 0.46     |      66.47 ± 0.51     |      23.91 ± 0.39     |
> |   ADML+A  | 66.02 ± 0.35 | 68.78 ± 0.37 |      24.46 ± 0.23  | 81.95 ± 0.38     |      67.97 ± 0.49     |      24.21 ± 0.28     |
> |   ADML+U  | 65.46 ± 0.40     |68.60 ± 0.33     | 24.58 ± 0.28 | 82.06 ± 0.36 | 68.21 ± 0.35}| 24.82 ± 0.34|
> |  ADML+A+U(new) |      64.24 ± 0.38     |   67.73 ± 0.45  |      23.88 ± 0.26     |      80.95 ± 0.41     |      67.64 ± 0.39     |      23.85 ± 0.30     |
>
> Table: ADML+A+U on SOP and INSHOP
>
> |           |     Online-product    |     |     |        In-shop        |   |   |
> |-----------|:---------------------:|:---------------------:|:---------------------:|:---------------------:|:---------------------:|:---------------------:|
> | Approachs |   R@1   |          NMI          |         mAP@C         |          R@1     |   NMI   | mAP@C   |
> |   ADML+T  |  77.13 ± 0.11 |   89.59 ± 0.03  | 40.75 ± 0.07  | 87.47 ± 0.12 |      89.65 ± 0.10 | 29.05 ± 0.12     |
> |   ADML+A  | 78.12 ± 0.16 |      89.95 ± 0.04     | 41.56 ± 0.11 | 87.94 ± 0.15| 89.93 ± 0.05| 30.12 ± 0.15|
> |   ADML+U  |      78.01 ± 0.12     | 89.97 ± 0.03 |      41.21 ± 0.12     |      87.86 ± 0.18  |      89.57 ± 0.08     |      29.93 ± 0.16     |
> |  ADML+A+U(new) |      77.41 ± 0.15     |      89.88 ± 0.07     |      40.91 ± 0.14 |      87.65 ± 0.12     |      89.71 ± 0.09     |      29.72 ± 0.13     |
>
>
> *2. ADML+Info-NCE model.*
>
> -- We provide an ablation study of ADML with different DML models, e.g. triplet, Info-NCE, and margin loss on CUB200-2011 (details in Sec  5.5 of the revision of our paper). ADML can boost the performance of Info-NCE loss, but it seems that the scale of improvement is similar to other metric losses (around 3\%). We also want to mention that in table 2 and 3 of our paper, the backbone DML for ADML is multi-similarity loss instead of triplet loss.
>
> Table: Recall@1 of different metric learning losses with ADML methods on CUB200-2011
>
> |                  | Vanilla | ADML+T | ADML+A | ADML+U | ADML+A+U |
> |------------------|:-------:|:------:|:------:|:------:|:--------:|
> | Triplet          |  62.29  |  63.68 |  65.11 |  64.72 |   63.17  |
> | Margin           |  62.48  |  64.26 |  65.92 |  65.61 |   64.32  |
> | Multi-similarity |  62.71  |  64.45 |  66.13 |  65.58 |   64.39  |
> | Info-NCE         |  61.42  |  62.74 |  64.01 |  63.85 |   62.91  |
>
> *3. Unfair comparison in Table 4*
>
> -- That's a good point, ADML+A definitely benefits from the alignment attacks. We will point this out in our paper.

---

> > ### Comment · Reviewer_MuXo · 2021-11-19
> > **Review Response**
> >
> > Thank  you for your rebuttal and feedback.
> > Tthe authors have addressed all the concerns I had in my initial review  through their revisions and comments. Moreover, as states by the authors in their other feedbacks, I believe that the theoretical and empirical contributions of this work are both novel and sufficiently supported. Therefore, I am raising my rating to an accept.

---

> > > ### Author Response · Authors · 2021-11-20
> > > **Thanks!**
> > >
> > > We are grateful to the reviewer for carefully reading and accepting our explanation and clarification. We also thank the reviewer for providing constructive comments.

---

### Official Review · Reviewer_UWyt · 2021-11-03

**Correctness:** 3
**Technical Novelty And Significance:** 2
**Empirical Novelty And Significance:** 2
**Recommendation:** 5
**Confidence:** 3

**Main Review:**


The paper is well written and easy to follow. Though the proposed idea is interesting and the theoretical results are solid, the claimed contributions are marginal given the related works [1,2]. To be specific,

1. I think the theoretical analysis that analyzes the intra-class alignment and hyperspherical uniformity for the tuple-based metric is a simple extension of [1] given the similarity between contrastive learning and deep metric learning (DML). In particular, the proof of main theoretical results in Section 3.2 in this paper inherits from the proof of Theorem 1 in [1] Appendix. Therefore, the theoretical contribution is marginal.

2. The idea of using adversarial samples for DML has been well explored by [2], though with a different attacking objective (triplet loss). As proved in this paper that the triplet loss also optimizes the intra-class alignment and hyperspherical uniformity goals, so the proposed method has a marginal contribution.

I have some other questions:
1. Proposition 1 is unclearly motivated. I cannot connect it with other analysis.

2. The claim that Page 3 “because the embedding space of DML is a unit hypersphere, where the maximum distance between two points is 2, it’s not possible to separate all negative embeddings with a large margin. Actually on $S^{k−1}$, the number of points with pairwise distance larger or equal than $\sqrt{2}$ is at most k and the embedding dimension k is always smaller than the number of feature vectors, thus it’s impossible to make all distances between negative pairs exceed $\sqrt{2}$” seems problematic. Assume k=2, then the number of points with a pairwise distance larger or equal than $\sqrt{2}$ is supposed to be 4, imaging four points of a square in the circle. (If I am wrong, please correct me.)

3. Why not consider a method ADML+A+U that includes two kinds of adversarial samples simultaneously?

4. What is the lambda for experiment section 5.5?

5. When increasing lambda, will the robustness of the model be better?

6. In equation (7), $L_{uniformity}$ should connect to equation (2) (3).



Reference

[1] Tongzhou Wang and Phillip Isola. Understanding contrastive representation learning through alignment and uniformity on the hypersphere. In International Conference on Machine Learning, pp.9929–9939. PMLR, 2020

[2] Yueqi Duan, Wenzhao Zheng, Xudong Lin, Jiwen Lu, and Jie Zhou. Deep adversarial metric learning. In Proceedings of the IEEE Conference on Computer Vision and Pattern Recognition, pp.
2780–2789, 2018.


**Summary Of The Paper:**

The paper connects metric losses with the intra-class alignment and hyperspherical uniformity from theoretical analysis and proposes to boost metric learning augmented with adversarial samples generated by attacking alignment or uniformity objective. Experiments compare the proposed method with various baselines and the proposed method achieves the best performance.

**Summary Of The Review:**

The paper has marginal contributions and limited novelty.

---

> ### Author Response · Authors · 2021-11-17
> **Author Response to Reviewer UWyt**
>
> Thanks for the reviewer's valuable comments.
>
> *Theoretical novelty compared to [1].*
>
> -- We agree that the intra-class alignment and hyperspherical uniformity are introduced by [1], and our work is also motivated by studying these two properties in DML. Although our proof steps in Theorem 2 follows [1], which splits the loss into positive and negative part, the proof techniques of our theorem is **totally different** from [1], which is our main theoretical contribution.
>
> *Emperically contribution compared to [2].*
>
> -- We respectively disagree that using adversarial samples for DML has been **well** explored by [2]. Firstly, the baseline in [2] has very poor performance. For example, the R@1 of triplet loss on CUB200-2011 is only 35.9, which is even worse than the image pretrained model (43.8) in Table 2 of our paper. We believe that the poor performance of the baseline can affect the credibility of their experimental results. Secondly, [2] didn't study the robust performance of DML models under adversarial attacks. Thirdly, [2] didn't study the effect of the weight of adversarial training $\lambda$ on the natural performance. In our work, we use a standardized DML training framework [3] and report the mean and standard deviation across 5 experiments, which enables fair comparison between the baseline and our proposed ADML models. We also show that the robustness of different ADML and DML models under alignment attacks, and show the ablation of $\lambda$ on the performance of our ADML models.
>
> Regarding other questions:
>
> *1. Proposition 1 is unclearly motivated.*
>
> -- In Proposition 1, we want to theoretically show that training with only intra-class alignment will result in mode collapse i.e. all samples will have the same embedding.
>
> *2. Problematic claim in page 3.*
>
> -- You are right, the number of points with pairwise distance larger or equal than $\sqrt{2}$ is at most 2k.
>
> *3. ADML+A+U*
>
> -- We include ADML+A+U in our experiments (we update Table 2 and Table 3 with ADML+A+U in the revision of our paper), but seems that it has worse performance than ADML+A or ADML+U, and similar performance as ADML+T.
>
> Table: ADML+A+U on CUB200-2011 and CARS196
>
> |           |      CUB200-2011      |    |      |        CARS196        |       |    |
> |-----------|:---------------------:|:---------------------:|:---------------------:|:---------------------:|:---------------------:|:---------------------:|
> | Approachs |  R@1    |     NMI       |  mAP@C   |   R@1   | NMI  | mAP@C   |
> |   ADML+T  | 64.37 ± 0.43     | 68.13 ± 0.49     |      24.05 ± 0.30|      80.88 ± 0.46 |      66.47 ± 0.51     |      23.91 ± 0.39     |
> |   ADML+A  | 66.02 ± 0.35 | 68.78 ± 0.37 |      24.46 ± 0.23  | 81.95 ± 0.38|      67.97 ± 0.49     |      24.21 ± 0.28     |
> |   ADML+U  | 65.46 ± 0.40     |68.60 ± 0.33| 24.58 ± 0.28 | 82.06 ± 0.36 | 68.21 ± 0.35}| 24.82 ± 0.34|
> |  ADML+A+U(new) |64.24 ± 0.38|   67.73 ± 0.45  |23.88 ± 0.26|      80.95 ± 0.41     |      67.64 ± 0.39     |      23.85 ± 0.30     |
>
> Table: ADML+A+U on SOP and INSHOP
>
> |           |     Online-product    |     |     |        In-shop        |   |   |
> |-----------|:---------------------:|:---------------------:|:---------------------:|:---------------------:|:---------------------:|:---------------------:|
> | Approachs |   R@1   |          NMI          |         mAP@C         |          R@1     |   NMI   | mAP@C   |
> |   ADML+T  |  77.13 ± 0.11 |   89.59 ± 0.03  | 40.75 ± 0.07  | 87.47 ± 0.12 |      89.65 ± 0.10 | 29.05 ± 0.12     |
> |   ADML+A  | 78.12 ± 0.16 |      89.95 ± 0.04     | 41.56 ± 0.11 | 87.94 ± 0.15| 89.93 ± 0.05| 30.12 ± 0.15|
> |   ADML+U  |      78.01 ± 0.12     | 89.97 ± 0.03 |      41.21 ± 0.12     |      87.86 ± 0.18  |      89.57 ± 0.08     |      29.93 ± 0.16     |
> |  ADML+A+U(new) |77.41 ± 0.15     |89.88 ± 0.07|40.91 ± 0.14 |      87.65 ± 0.12     |      89.71 ± 0.09     |      29.72 ± 0.13     |
>
>
> *4. $\lambda$ for experiment section 5.5*
>
> -- Our training objective is $L=L_{vanilla}+\lambda L_{adv}$, thus $\lambda$ is the weight of the adversarial training in our learning objective $L$.
>
> *5. Effect of $\lambda$ on robustness.*
>
> --
> Yes, increasing $\lambda$ will result in better robustness of the DML model. we will put the results in our paper (see Sec. 5.5).
>
> Table: Robustness with different adversarial training strength $\lambda$ on CUB200-2011 under the alignment attacks. The metric is Recall@1.
>
> | $\lambda$ |   0  |  0.2  |  0.4  |  0.6  |  0.8  |  1.0  |
> |-----------|:----:|:-----:|:-----:|:-----:|:-----:|:-----:|
> | ADML+A    | 8.90 | 19.15 | 22.51 | 24.47 | 25.29 | 25.93 |
> | ADML+U    | 8.90 | 16.21 | 18.36 | 20.23 | 21.06 | 21.55 |
>
> *6. equation (7) should connect to equation (2) (3)*
>
> -- We will add the connection between equation (7) and equations (2) (3)
>
> [1] Wang et al. Understanding Contrastive Representation Learning through Alignment and Uniformity on the Hypersphere, ICML 2020
>
> [2] Duan et al. Deep Adversarial Metric Learning. CVPR 2018.

---

> ### Author Response · Authors · 2021-12-02
> **Thank you again for the review and hope the reviewer can check out our response**
>
> Dear Reviewer UWyt,
>
> Thank you again for the valuable comments. We hope the reviewer can read our response and reevaluate our paper based on our response and the revised paper. Please let us know if you have further questions about our paper and we look forward to hearing from you.
>
> Sincerely, Paper1943 Authors.

---

### Official Review · Reviewer_t4QZ · 2021-11-05

**Correctness:** 3
**Technical Novelty And Significance:** 2
**Empirical Novelty And Significance:** 3
**Recommendation:** 6
**Confidence:** 3

**Main Review:**

I like the general idea of the paper and the hyperspherical perspective on deep metric learning. It has been a common wisdom that deep metric learning should be performed on a hyperspherical space, but an rigorous and principled understanding is yet to be found. I believe this is an interesting and extremely important direction. In such sense, this paper is tackling an significant problem.

The proposed algorithm to improve deep metric learning method is rather straightforward. Instead of generating the adversarial examples using the standard metric learning loss (e.g. triplet loss), the paper takes another approach by attacking either intra-class alignment loss or hyperspherical uniformity loss. I think the intuition behind makes senses, and the method can be viewed as attacking the deep metric learning objective from a dual space (which is equivalent to tuple-based loss space under some regularity).

The empirical results seem good to me. The proposed ADML by attacking intra-class alignment loss / hyperspherical uniformity loss shows non-trivial gain compared to the other existing methods.

My concerns and suggestions are mostly in the following aspects:

(1) I understand that the decompostion to intra-class alignment and hyperspherical uniformity is from tuple-based losses, but will the generated adversarial examples also help contrastive loss / triplet loss? It will be interesting to see whether these adversarial examples are generally useful for deep metric learning methods.

(2) Since attacking either intra-class alignment loss or hyperspherical uniformity loss yields a consistent performance gain, what if attack both objective at the same time? It will be more interesting to formulate the attack to both intra-class alignment loss and hyperspherical uniformity loss in a soft manner. That is to say, you have a parameter to control the balance between attacking two losses. Although you end up with an additioanl hyperparameter, it will make the method more flexible and I will be curious to see how the performance varies with this hyperparameter.

(3) It will strengthen the experiments if there could be comparison among different adversarial attack methods. If every adversarial attack method works well for improving tuple-based losses, it will make the current conclusion more general.

(4) The presentation could be significantly improved and I think the current paper structure may be suboptimal. I think some of the analyses on decomposing tuple-based loss to intra-class aligment and hyperspherical uniformity can be put to appendix. Only the main result is put in the main paper. Then the authors can be sufficient space to comprehensively evaluate how and why these generated adversarial examples help deep metric learning.

(5) It will be helpful to visualize these adversarial examples on a 2-dim embedding space. Specifically, you can train the same model by setting the output feature dimension as 2. Then these features can be naturally visualized without any other visualization tools (such as T-SNE). Of course, using T-SNE may also be fine and interesting to see where these examples lie geometrically.

I am sitting between weak accept and accept. If my concerns are properly addressed, I will be happy to increase my score.

**Summary Of The Paper:**

This paper first analyzes the tuple-based metric losses on the hyperspherical space and studies two objective functions: intra-class alignment and hyperspherical uniformity. Then based on these two objective functions, the paper develops two ways of generating adversarial examples for improving deep metric learning. The empirical results show some advantages of the proposed method.

**Summary Of The Review:**

This paper studies an interesting problem. Although the proposed method seems naive and incremental, its effectiveness and the significance of the topic really make up for that. Overall, I vote for weak accept, but may increase my rating based on the rebuttal.

---

> ### Author Response · Authors · 2021-11-17
> **Author Response to Reviewer t4QZ**
>
> Thanks for the reviewer's valuable comments.
>
> (1) Yes, we provide an ablation study of ADML with different DML models e.g. triplet, Info-NCE, and margin loss on CUB200-2011 (details in Sec. 5.5 of the revision of our paper). Our ADML methods can consistently boost the performance of these methods.
>
> Table: Recall@1 of different metric learning losses with ADML methods on CUB200-2011
>
> |      | Vanilla | ADML+T | ADML+A | ADML+U | ADML+A+U |
> |------------------|:-------:|:------:|:------:|:------:|:--------:|
> | Triplet  |  62.29  |  63.68 |  65.11 |  64.72 |   63.17  |
> | Margin   |  62.48  |  64.26 |  65.92 |  65.61 |   64.32  |
> | Multi-similarity |  62.71  |  64.45 |  66.13 |  65.58 |   64.39  |
> | Info-NCE         |  61.42  |  62.74 |  64.01 |  63.85 |   62.91  |
>
> (2) We did experiments on attacking both alignment and uniformity objective (ADML+A+U) on all four benchmarks (we update Table 2 and Table 3 with ADML+A+U in the revision of our paper), and formulate the ADML with A+U attack in a soft manner (ADML+(1-$\beta$)A+$\beta$U) on CUB200-2011 (details in Sec 5.5 of the revision of our paper). When $\beta$ is increased, the Recall@1 of ADML model  decreases first and then increases, which indicates that attacking alignment and uniformity loss separately can lead to better results
>
> Table: ADML+A+U on CUB200-2011 and CARS196
>
> |           |      CUB200-2011      |         |           |        CARS196        |           |     |
> |-----------|:---------------------:|:---------------------:|:---------------------:|:---------------------:|:---------------------:|:---------------------:|
> | Approachs | R@1 |  NMI   |         mAP@C    |          R@1          |          NMI          |  mAP@C   |
> |   ADML+T  |      64.37 ± 0.43     |      68.13 ± 0.49     |      24.05 ± 0.30     |      80.88 ± 0.46     |      66.47 ± 0.51     |      23.91 ± 0.39     |
> |   ADML+A  | 66.02 ± 0.35 | 68.78 ± 0.37 |      24.46 ± 0.23|      81.95 ± 0.38     |      67.97 ± 0.49     |      24.21 ± 0.28     |
> |   ADML+U  | 65.46 ± 0.40     |      68.60 ± 0.33     | 24.58 ± 0.28| 82.06 ± 0.36 |68.21 ± 0.35 | 24.82 ± 0.34|
> |  ADML+A+U(new) | 64.24 ± 0.38|67.73 ± 0.45|23.88 ± 0.26|80.95 ± 0.41|67.64 ± 0.39     |      23.85 ± 0.30     |
>
> Table: ADML+A+U on SOP and INSHOP
>
> |           |     Online-product    |      |          |        In-shop        |      |      |
> |-----------|:---------------------:|:---------------------:|:---------------------:|:---------------------:|:---------------------:|:---------------------:|
> | Approachs |     R@1          |     NMI |   mAP@C  |  R@1 |    NMI          |         mAP@C         |
> |   ADML+T  | 77.13 ± 0.11|      89.59 ± 0.03     |      40.75 ± 0.07     |      87.47 ± 0.12     |      89.65 ± 0.10     |      29.05 ± 0.12     |
> |   ADML+A  | 78.12 ± 0.16 |      89.95 ± 0.04     | 41.56 ± 0.11 | 87.94 ± 0.15 | 89.93 ± 0.05 | 30.12 ± 0.15 |
> |   ADML+U  | 78.01 ± 0.12| 89.97 ± 0.03 |      41.21 ± 0.12     |      87.86 ± 0.18     |      89.57 ± 0.08     |      29.93 ± 0.16     |
> |  ADML+A+U(new) | 77.41 ± 0.15| 89.88 ± 0.07     |40.91 ± 0.14  |      87.65 ± 0.12     |      89.71 ± 0.09     |      29.72 ± 0.13     |
>
> Table: Recall@1 of ADML+$(1-\beta)$A+$\beta$ U on CUB200-2011 with different $\beta$.
>
> | $\beta$       |   0   |  0.2  |  0.4  |  0.5  |  0.6  |  0.8  |   1   |
> |-------------------------------|:-----:|:-----:|:-----:|:-----:|:-----:|:-----:|:-----:|
> | ADML+$(1-\beta)$A+$(\beta)$ U | 66.13 | 65.81 | 64.67 | 64.39 | 64.55 | 65.33 | 65.58 |
>
> (3)  It's a nice idea, however, to our best knowledge the adversarial attack on metric learning tasks is mainly triplet based [1][2][3] (could have minor difference based on the specific tasks).  We don't have other kinds of metric learning attacks available.
>
> (4)  You are right. We put the discussion of naive linear loss into the appendix, and move our new experimental results into the main paper (Sec 5.5).
>
> (5) *... setting the output feature dimension as 2.*
>
> -- This is a good idea, but DML with only 2 feature dimension suffers very poor performance. We have to keep moderate large dimension to represent the features of the images.
>
> *Tsne results.*
>
> -- We add the T-SNE results with the normal embedding size (128) in the experiments (figures in Appendix F of the revision of our paper). We first generate the 128-dim feature vectors, and then train a T-SNE model with the feature vectors and select the first 10 classes of images to visualize. The TSNE plots show that the ADML model can better separate the images of different classes. For further details, please see the revision of our paper. We don't put it in the main paper because of the limitation of space.
>
> [1] Wang et al. Transferable, controllable, and inconspicuous adversarial attacks on person re-identification with deep mis-ranking. CVPR 2020
>
> [2] Zhou et al. Qilin Zhang, and GangHua.  Adversarial ranking attack and defense. ECCV 2020.
>
> [3] Panum et al. Exploring Adversarial Robustness of Deep Metric Learning.

---

> ### Author Response · Authors · 2021-12-02
> **Thank you again for the review and hope the reviewer can check out our response**
>
> Dear Reviewer t4QZ,
>
> Thank you again for the valuable comments. We hope the reviewer can read our response and reevaluate our paper based on our response and the revised paper. Please let us know if you have further questions about our paper and we look forward to hearing from you.
>
> Sincerely, Paper1943 Authors.

---

### Official Review · Reviewer_xCnR · 2021-11-06

**Correctness:** 3
**Technical Novelty And Significance:** 3
**Empirical Novelty And Significance:** 3
**Recommendation:** 5
**Confidence:** 4

**Details Of Ethics Concerns:**

The paper shows a way to produce adversarial examples that can affect privacy, security, and safety by one or more ways.

**Main Review:**

Strengths
--------------------------------------------
1. Fairly well written, though a few sections, such as Sec 2 and 4 can be improved
2. Theoretical proves are given however they are largely motivate by [1]

Weakness
-------------------------------------------
1. Why the unit hypersphere is a nice feature space is not very well addressed in this paper. Is there any link between connected
sets with smooth boundaries are nearly linearly separable in the hyperspherical geometry and such linear separability is used here to disentangle relation?
2. "... We derive the theoretical analysis for the triplet loss to prove that the triplet loss..." Page 1, Para 2. IMHO this fact is already well studied and theoretically proved in [1]. A citation to [1] is missing on Page 1, Para 2.
3. Ablation missing? For example, a study considering only positive, only contrastive, etc., is missing?
4. Negative samples are hard negative or soft negatives? How are we getting negatives is not very clear.

Ref:
-----------------------------------------------------
1. Understanding Contrastive Representation Learning through Alignment and Uniformity on the Hypersphere, ICML 2020

**Summary Of The Paper:**

In this paper, the authors provided elegant similarities between contrastive learning and deep metric learning. They decomposed contrastive loss objectives into two quantities evaluating the geometry of the learned representation space:
1. Alignment, and,
2. Uniformity. Where Alignment is the closeness that defines how close two positive pairs are on the embedding space. While the uniformity ensures the scattered. The authors found the representation learned by optimizing contrastive objective indeed has these two properties compared to representation learned using supervised objective. They also showed that both alignment and uniformity are required for learning better examples for adversarial example tasks.

**Summary Of The Review:**

The paper shows a closeness between contrastive learning and deep metric learning. They decomposed contrastive loss objectives into two quantities evaluating the geometry of the learned representation space:
1. Alignment, and,
2. Uniformity. However, the paper misses producing few ablations such as considering only positive, only contrastive, etc. How we mine negative samples is not very clear. Overall the paper seems like an extension of [1].


Ref:
-----------------------------------------------------
1. Understanding Contrastive Representation Learning through Alignment and Uniformity on the Hypersphere, ICML 2020

---

> ### Author Response · Authors · 2021-11-17
> **Author Response to Reviewer xCnR**
>
> Thanks for the reviewer's valuable comments.
>
> *1. Why the unit hypersphere is a nice feature space?*
>
> -- The advantage of unit hypersphere feature space has been well addressed in previous works [3-5]. Compared to the $\mathbb{R}^d$ embedding space, unit hypersphere embedding can help prevent overfitting and improve the generalization of neural works [3].
>
> *2. Is there any link between connected sets with smooth boundaries...*
>
> -- You are right, the linear separability of unit hypersphere has been discussed in [1], and this property is particularly good for contrastive representation learning. Because in contrastive learning, we train a linear classifier to separate the feature vectors on the unit hypersphere.
>
> *3. A citation to [1] is missing on Page 1, Para 2.8*
>
> -- We will cite [1] on Page 1, Para 2. We hope to point out that [1] only proved the contrastive learning objective i.e. Info-NCE loss is related to intra-class alignment and hyperspherical uniformity. We prove the triplet loss case, where the proof technique is \textbf{completely different} from the Info-NCE loss.
>
> *4. Ablation missing and negative sampling*
>
> -- we provide an ablation study of ADML with different DML models e.g. triplet, Info-NCE, and margin loss on CUB200-2011 (details in Sec. 5.5 of the revision of our paper). We generated negatives following the settings of DML models in [2]. For examples, we use distance weighted sampling as the negative miner for DML with margin loss, and we use the Similarity-P tuple mining for multi-similarity loss.
>
> Table: Recall@1 of different metric learning losses with ADML methods on CUB200-2011
>
> |                  | Vanilla | ADML+T | ADML+A | ADML+U | ADML+A+U |
> |------------------|:-------:|:------:|:------:|:------:|:--------:|
> | Triplet          |  62.29  |  63.68 |  65.11 |  64.72 |   63.17  |
> | Margin           |  62.48  |  64.26 |  65.92 |  65.61 |   64.32  |
> | Multi-similarity |  62.71  |  64.45 |  66.13 |  65.58 |   64.39  |
> | Info-NCE         |  61.42  |  62.74 |  64.01 |  63.85 |   62.91  |
>
> *Our contributions*
>
> We want to emphasize our work is not an extension of [1]. We agree that the alignment and uniformity objectives are motivated by [1], however, a) our proof technique for triplet loss is different from [1]; b) our main contribution is designing ADML approaches which can improve the performance and robustness of DML models simultaneously. This is a novel finding that is not discussed in [1]. Thus we disagree that our work is an extension of [1].
>
> [1] Wang et al. Understanding Contrastive Representation Learning through Alignment and Uniformity on the Hypersphere, ICML 2020
>
> [2] Roth et al. Revisiting training strategies and generalization performance in deep metric learning, ICML 2020.
>
> [3] Liu et al.  Learning with hyperspherical uniformity. AISTATS 2021.
>
> [4] Liu et al. Learning towards minimum hyperspherical energy. 	NeurIPS 2018.
>
> [5] Liu et al. Sphereface:  Deep hypersphere embedding for face recognition. CVPR 2017.

---

> ### Author Response · Authors · 2021-12-02
> **Thank you again for the review and hope the reviewer can check out our response**
>
> Dear Reviewer xCnR,
>
> Thank you again for the valuable comments. We hope the reviewer can read our response and reevaluate our paper based on our response and the revised paper. Please let us know if you have further questions about our paper and we look forward to hearing from you.
>
> Sincerely, Paper1943 Authors.

---

### Author Response · Authors · 2021-11-20
**Shared Reply to All Reviewers**

We would like to thank all reviewers for their thoughtful comments and constructive suggestions. In the revised paper we move our theoretical discussion about naive linear metric losses to Appendix B, and add ablation studies about

(1) effect of adversarial training strength $\lambda$ on the robustness of ADML (Table 5).

(2) ADML on different metric loss (Table 6).

(3) a mixture of alignment and uniformity attacks (Table 7)

to the Sec 5.5 of our paper.

We also add the T-SNE results of the embedding generated by vanilla DML and ADML+A in Appendix F. We mark the changed parts in the revised paper with red for convenience.

---

### Decision · Program_Chairs · 2022-01-20

**Decision:**

Reject

**Comment:**

While the paper has merits, I generally agree with negative reviewers. Among other issues, there were concerns about the theoretical contribution overlaps with prior work. While the authors argued the current work is not an extension, but rather designing ADML is. If this is the case, the paper should be rewritten to deemphasize the less novel contribution and focus more on what the authors believe to be the novel contribution. I don't believe in the practice of putting different messages (some novel and some not) into a paper with the hope that this makes the overall result "more novel". I'd suggest the author rewrite the paper and more clear about the message.